# Federated Causal Discovery with Additive Noise Models

## Abstract

Causal discovery aims to learn a causal graph from observational data. To date, most causal discovery methods require data to be stored in a central server. However, data owners gradually refuse to share their personalized data to avoid privacy leakage, making this task more troublesome by cutting off the first step. A puzzle arises: *how do we infer causal relations from decentralized data?* In this paper, focusing on the additive noise models (ANMs) assumption of data, we take the first step in developing a gradient-based learning framework named DAG-Shared Federated Causal Discovery (DS-FCD), which can learn the causal graph without directly touching the local data and naturally handle the data heterogeneity caused by causal mechanism or noise shift. DS-FCD benefits from a two-level structure of each local model. The first level structure learns the causal graph and communicates with the server to get the model information from other clients during the learning procedure, while the second level structure approximates the causal mechanisms and personally updates from its own data to accommodate the data heterogeneity. Moreover, DS-FCD formulates the overall learning task as a continuous optimization problem by taking advantage of an equality acyclicity constraint, which can be solved by gradient descent methods. Extensive experiments on both synthetic and real-world datasets verify the efficacy of the proposed method.

## 1 Introduction

The discovery of causal relations among concerned variables is a fundamental and challenging problem in various fields, such as econometrics (Heckman, 2008), epidemiology (Greenland et al., 1999), and biological sciences (Imbens & Rubin, 2015). The requirement comes from the need of excavating the generation process behind data, guiding actions and policies, learning from the past (Pearl et al., 2016). To achieve this goal, a reliable way is to conduct randomized controlled (control) trials, which, however, may face difficulty or even be ethically forbidden in some cases (Resnik, 2008; Nardini, 2014). By leveraging the use of directed acyclic graphs (DAGs) to represent the cause-effect relations among variables, causal discovery, which directly infers the causal relations from observational data by learning a DAG, brings a new solution to this problem and has received a great deal of attention (Peters et al., 2017; Glymour et al., 2019).

Various methods (Spirtes et al., 2001; Chickering, 2002; Shimizu et al., 2006; Zheng, 2020) for learning causal relations from purely observational data have been proposed over the recent decades. In practice, however, finite sample problem bears the brunt of performance decrease of causal discovery method. Regularly, (1) collecting data from various sources and then (2) designing a causal discovery algorithm on all collected data can serve as a straightforward and common pipeline to alleviate this issue in this field. However, owing to the issue of data privacy, data owners gradually prefer not to share their personalized data[1] with others (Kairouz et al., 2021). Naturally, the new predicament, *how do we infer causal relations from decentralized data?* has arisen. In statistical learning problems such as regression and classification, federated learning (FL) has been proposed to learn from locally stored data (McMahan et al., 2017). Inspired by the developments in FL, we aim to develop a federated causal discovery (FCD) framework that enables to learn DAG from decentralized data. Compared to the traditional FL methods in statistical learning, FCD, a **structural learning** task, has the following two main differences:

---

[1]In this paper, we restrict our scope to define the privacy leakage by sharing the raw data of users.

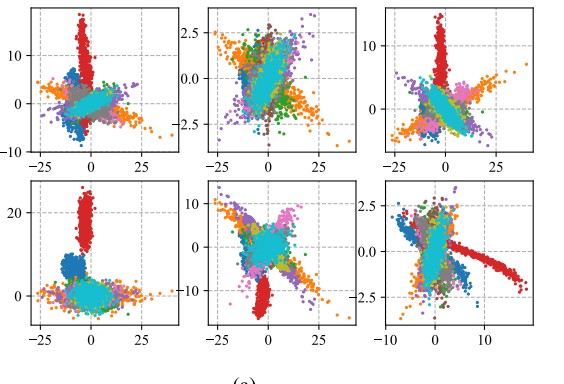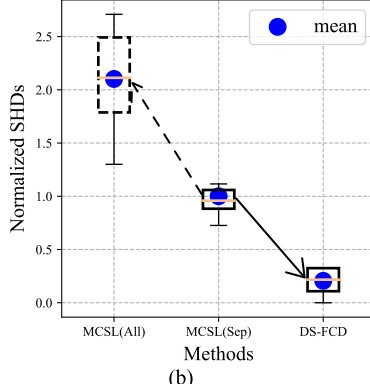

(a)                         (b)

Figure 1: (a) Visualization of heterogeneous data, where different colors represent data from different sources while each sub-figure includes the distribution of one fixed dimension of data from all clients. (b) Normalized structural hamming distances (SHDs) (↓) of three methods, where MCSL (Sep) (Ng et al., 2022b) separately trains model on local data while MCSL (All) trains one model on all data, which, however, is forbidden in FL.

- **Learning objective difference.** Most of the previous FL researches focus on learning *an estimator* to estimate the conditional distribution $P(Y|X)$ in supervised learning tasks, e.g., image classification (McMahan et al., 2017), sequence tagging (Lin et al., 2022), and feature prediction (Kairouz et al., 2021). However, FCD, an unsupervised learning task, tries to *find the underlying causal structure* among the concerned variables and the *causal mechanisms estimator* to fit with the joint distribution of observations.

- **Data heterogeneity difference.** FL mainly cares data heterogeneity, which is assumed to be caused by some specific distribution shift types such as label shift (the shift of $P(Y)$) Lipton et al. (2018) or covariate shift (the shift of $P(X)$) Reisizadeh et al. (2020), while FCD handles a generative model where data heterogeneity can admit the joint distribution shift of all variables (as shown in Figure 1(a)), which would bring more challenges compared to the model design in the federated learning paradigm.

To overcome the aforementioned problem, we present DAG-Shared Federated Causal Discovery (DS-FCD), a gradient-based framework for learning the underlying causal graph and causal mechanisms from decentralized data, including the case of heterogeneous data caused by causal mechanism or noise shift. (1) To alleviate the data leakage problem, DS-FCD inherits the merits of FL, which proposes to separately deploy a local model to each client and collaboratively learn a joint model at the server end. Instead of sharing raw data, DS-FCD exchanges model-info among clients and the server to achieve collaboration. (2) Taking into consideration of the first main difference between FCD and FL, a two-level structure consisting of causal graph learning (CGL) part and causal mechanisms approximating (CMA) part respectively, is adopted as the local model. (3) Benefiting from this separated structure, the second difference between FL and FCD can naturally be handled by only sharing CGL parts of clients during FL and locally updating CMA to get with data heterogeneity. Moreover, we provide the identifiability conditions for learning the causal graph from decentralized data. Our contributions are summarized as follows:

- We introduce FCD, under the assumption that the underlying causal graph among different datasets remains invariant, while causal mechanisms and noises distributions are allowed to vary. We also show the identifiability conditions of causal discovery from decentralized data.

- We propose DS-FCD, which separately learns the causal mechanisms on local data and jointly learns the causal graph to elegantly handle data heterogeneity. Meanwhile, since 0 bits of raw data is shared but only parameters of the CGL parts of models, the requirement of the privacy protection is guaranteed and the communication pressure is quite low.

- We evaluate our proposed method with data following an SEM with an additive noise structure on a variety of experimental settings, including simulations and real dataset, against recent state-of-the-art algorithms for showing its superior performance and the ability to use one model for all settings.

## 1.1 Potential applications of FCD.

Compared to traditional causal discovery methods, the IID setting of our FCD just brings one more assumption that local data cannot be directly collected owing to the consideration of *privacy leakage*. Then, we further extend our model to heterogeneous data, where causal mechanism and noise distributions may also vary among different local data. Therefore, our method could be directly applied to the applications of causal discovery where privacy is also very important.

The first example can come from medical science. Exploring causal relations from healthcare data can help to understand the disease mechanisms and causes (Yang et al., 2013). However, in real medical scenarios, the clinical data of patients are extremely sensitive and absolutely related to personal privacy, which faces very strict data protection regulations, such as HIPAA regulations (Annas, 2003). For some rare diseases, each hospital may own finite clinical data, which, however, is not enough for causal discovery. How can hospitals cooperate to analyze the pathology while preventing sharing the privacy information (raw diagnostic data)? Naturally, this challenge can be addressed by our method. Depending on how each hospital collects the data, e.g., medical devices, and survey design, the data in each hospital may not share the same distribution.

The second example can come from the recommendation system (RS) (Wang et al., 2020b; Yang et al., 2020). Introducing causal model into RS is becoming prevail since leveraging causality can perform robust recommendation by de-confounding some spurious relations. As users pay more attention to privacy and governments also exacerbate many strict regulations like the General Data Protection Regulation (GDPR), it brings increasing difficulties to collect the personal raw data to the server. Accordingly, we think that RS can also benefit from FCD.

## 2 Preliminaries

**Additive Noise Models (ANMs).** We consider a specific structural causal model (SCM), which is defined as a triple $\mathcal{M} = \langle \mathcal{X}, \mathcal{E}, \mathcal{F} \rangle$, where $\mathcal{E}$ is a set $\{\epsilon_1, \epsilon_2, \cdots, \epsilon_d\}$ of exogenous variables and $\mathcal{X} = \{X_1, X_2, \cdots, X_d\}$ is a set of endogenous variables. $\mathcal{F} = \{f_1, f_2, \cdots, f_d\}$ is a set of functions, where each $f_i$, called the causal mechanism of $X_i$, maps $\epsilon_i \cup \mathbf{PA}_i$ to $X_i$, i.e., $X_i = f_i(\mathbf{PA}_i, \epsilon_i)$, where the $\mathbf{PA}_i$ corresponds to the set including all direct parents of $X_i$. $\mathcal{M}$ can be leveraged to describe how nature assigns values to variables of interest (Pearl et al., 2016). In this paper, we narrow our focus to a commonly used model named ANMs. They assume that

$$X_i = f_i(\mathbf{PA}_i) + \epsilon_i, \quad i = 1, 2, \cdots, d, \tag{1}$$

where $\epsilon_i$ is always taken as a random noise, which is independent of variables in $\mathbf{PA}_i$ and mutually independent with any $\epsilon_j$ for $i \neq j$.

**Probabilistic Causal Graphical Models (PCGM).** Let $X = (X_1, X_2, \cdots, X_d)$ be a vector that includes all variables in $\mathcal{X}$ with index set $\mathbb{V} := \{1, 2, \cdots, d\}$ and $P(X)$ with the probability density function $p(X)$ be a marginal distribution induced from $\mathcal{M}$. A DAG $\mathcal{G} = (\mathbb{V}, \mathbb{E})$ consists of a nodes set $\mathbb{V}$ and an edge set $\mathbb{E} \subseteq \mathbb{V}^2$. Every causal model $\mathcal{M}$ can be associated with a DAG $\mathcal{G}_\mathcal{M}$, in which each node $i$ corresponds to the variable $X_i$ and directed edges point from $\mathbf{PA}_i$ to $X_i$[2] for $i \in [d]$[3]. A PCGM is defined as a pair $\langle P(X), \mathcal{G}_\mathcal{M} \rangle$. Then $\mathcal{G}_\mathcal{M}$ is called the causal graph associated with $\mathcal{M}$ and $P(X)$ is Markovian to $\mathcal{G}_\mathcal{M}$. Throughout the main text, we assume the *causal sufficiency condition*[4] (no hidden variable) (Spirtes et al., 2001) and then $p(X)$ can be factorized as

$$p(X) = \prod_{i=1}^{d} p(X_i | X_{pa_i}) \tag{2}$$

according to $\mathcal{G}_\mathcal{M}$ (Lauritzen, 1996). $X_{pa_i}$ is the parental vector that includes all variables in $\mathbf{PA}_i$.

**Characterizations of Acyclicity.** A DAG $\mathcal{G}$ with $d$ nodes can be represented by a binary adjacency matrix $\boldsymbol{B} = [\boldsymbol{B}_{:,1} | \boldsymbol{B}_{:,2} | \cdots | \boldsymbol{B}_{:,d}]$ with $\boldsymbol{B}_{:,i} \in \{0,1\}^d$ for $\forall i \in [d]$. NOTEARS (Zheng et al., 2018) first formulates a

---

[2]In the intact causal graph of ANMs, we just fix directed edges from $\epsilon_i$ to $X_i$ and assume the distribution of $\epsilon_i$. Therefore, in this paper, $\mathcal{G}$ is only defined over the endogenous variables.

[3]For simplicity, we use $[d] = \{1, 2, \cdots, d\}$ to represent the set of all integers from 1 to $d$.

[4]This assumption can be relaxed to some restricted cases with hidden variables. See Appendix C.4 for details.

sufficient and necessary condition for $\boldsymbol{B}$ representing a DAG by an equality constraint. The formulation is as follows:

$$\text{Tr}[e^{\boldsymbol{B}}] - d = 0, \tag{3}$$

where $\text{Tr}[\cdot]$ means the trace of a given matrix. $e^{(\cdot)}$, here, is the matrix exponential operation. However, NOTEARS is only designed to solve the linear Guassian models, which assume that all the causal mechanisms are linear. Therefore, the causal graph and causal mechanisms can be modeled together by a weighted matrix. To extend NOTEARS to the non-linear cases, MCSL (Ng et al., 2022b) proposes to use a mask $\boldsymbol{M}$, parameterized by a continuous proxy matrix $\boldsymbol{U}$, to approximate the adjacency matrix $\boldsymbol{B}$. To enforce the entries of $\boldsymbol{M}$ to approximate the binary form, i.e., 0 or 1, a two-dimensional version of Gumbel-Softmax (Jang et al., 2017) approach named Gumbel-Sigmoid is designed to reparameterize $\boldsymbol{U}$ and to ensure the differentiability of the model. Then, $\boldsymbol{M}$ can be obtained element-wisely by

$$\boldsymbol{M}_{ij} = \frac{1}{1 + \exp(-\log(\boldsymbol{U}_{ij} + \text{Gumb}_{ij})/\tau)}, \tag{4}$$

where $\tau$ is the temperature, $\text{Gumb}_{ij} = g_{ij}^1 - g_{ij}^0$, $g_{ij}^1$ and $g_{ij}^0$ are two independent samples from $\text{Gumbel}(0,1)$. For simplicity but equivalence, $g_{ij}^1$ and $g_{ij}^0$ also can be sampled from $-\log(\log(a))$ with $a \sim \text{Uniform}(0,1)$. See Appendix D in (Ng et al., 2022b). MCSL names Eq. (4) as Gumbel-Sigmoid w.r.t. $\boldsymbol{U}$ and temperature $\tau$, which is written as $g_\tau(\boldsymbol{U})$. Then, the acyclicity constraint can be reformulated as

$$\text{Tr}[e^{(g_\tau(\boldsymbol{U}))}] - d = 0. \tag{5}$$

## 3    Problem definition

Here, we first describe the property of decentralized data and the data distribution shift among different clients if there exists data heterogeneity (Huang et al., 2020b; Mooij et al., 2020; Zhang et al., 2020). Then, we define the problem, federated causal discovery, considered in this paper.

**Decentralized Data and Probability distribution set.** Let $\mathcal{C} = \{c_1, c_2, \cdots, c_m\}$ be the client set which includes $m$ different clients and $s$ be the only server. The data $\mathcal{D}^{c_k} \in \mathbb{R}^{n_{c_k} \times d}$, in which each observation $\mathcal{D}_i^{c_k}$ for $\forall i \in [n_{c_k}]$ independently sampled from its corresponding probability distribution $P^{c_k}(X)$, represents the personalized data owned by the client $c_k$. $n_{c_k}$ is the number of observations in $\mathcal{D}^{c_k}$. The dataset $\mathcal{D} = \{\mathcal{D}^{c_1}, \mathcal{D}^{c_2}, \cdots, \mathcal{D}^{c_m}\}$ is called a decentralized dataset and $P^{\mathcal{C}}(X) = \{P^{c_1}(X), P^{c_2}(X), \cdots, P^{c_m}(X)\}$ is defined as the decentralized probability distribution set. If $P^{c_{k_1}}(X) = P^{c_{k_2}}(X)$ for $\forall k_1, k_2 \in [m]$, then $\mathcal{D}$ is defined as an independent and identically distributed (IID) decentralized dataset throughout this paper. The heterogeneous decentralized dataset is defined by assuming that there exists at least two clients, e.g., $c_{k_1}$ and $c_{k_2}$, on which the local data are sampled from different distributions, i.e., $P^{c_{k_1}}(X) \neq P^{c_{k_2}}(X)$.

**Assumption 3.1. (Invariant DAG)** For $\forall c_k$, $P^{c_k}(X) \in P^{\mathcal{C}}(X)$ admits the product factorization of Eq. (2) relative to the same DAG $\mathcal{G}$.

*Remark* 3.2. If $P^{\mathcal{C}}(X)$ satisfies Assumption 3.1, then, each $P^{c_k}(X) \in P^{\mathcal{C}}(X)$ is Markov relative to $\mathcal{G}$.

According to the general definition of *mechanism change* in (Tian & Pearl, 2001), interventions can be seen as a special case of distribution changes, where the external influence involves fixing certain variables to some predetermined values. Actually, in general, the external influence may be milder to just merely change the conditional probability of certain variables given its causes. In this paper, we restrict our scope by assuming that distribution shift across $P^{c_k}(X)$ comes from the changes of causal mechanisms in $\mathcal{F}$ or distributions shift of the exogenous variables in $\mathcal{E}$ (see Appendix F.1 for detailed discussion).

**Assumption 3.3.** For $\forall c_{k_1}, c_{k_2}$, if $P^{c_{k_1}}(X) \neq P^{c_{k_2}}(X)$, the distribution shift is caused by (1) $\exists\ i \in [d]$, $P^{c_{k_1}}(X_i|X_{pa_i}) \neq P^{c_{k_2}}(X_i|X_{pa_i})$, i.e., $f_i^{c_{k_1}} \neq f_i^{c_{k_2}}$ (2) $\exists\ i \in [d]$, $P^{c_{k_1}}(\epsilon_i) \neq P^{c_{k_2}}(\epsilon_i)$.

**Federated Causal Discovery.** Given the decentralized dataset $\mathcal{D}$ consisting of data from $m$ clients while the corresponding $P^{\mathcal{C}}(X)$ satisfies Assumptions 3.1 and 3.3, the aim of federated causal discovery is to identify the underlying DAG $\mathcal{G}$ from $\mathcal{D}$.

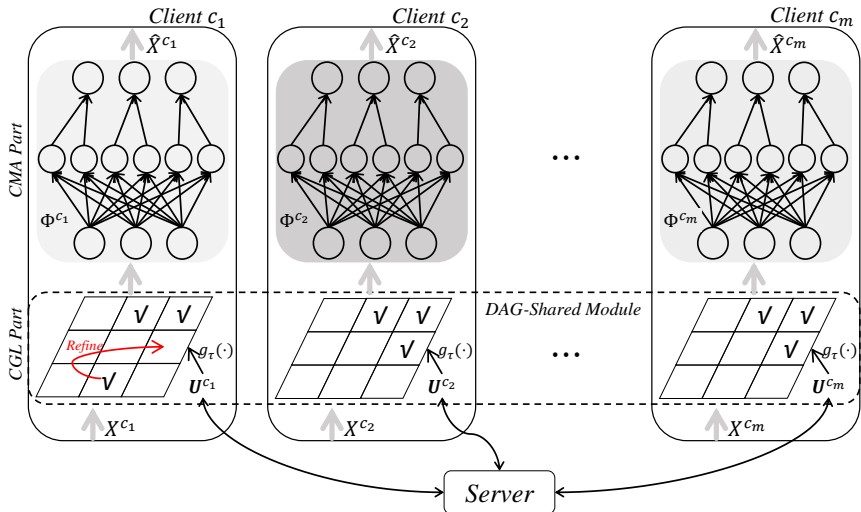

Figure 2: An overview of DS-FCD. Each solid-line box includes the local model for each client. For client $c_k$, the CGL part includes a continuous proxy $U^{c_k}$ and $g_\tau(\cdot)$, the Gumbel-Sigmoid function, which maps $U^{c_k}$ to approximate the binary adjacency matrix. The CMA part uses $\Phi^{c_k}$, including $d$ neural networks, to approximate the causal mechanisms. $X^{c_k}$ represents observations on $c_k$ and $\hat{X}^{c_k}$ is the predicted data. $X^{c_k}$ firstly goes through the CGL part to select the parental variables and then the CMA part to get $\hat{X}^{c_k}$. The server coordinates the FL procedures by leveraging $U$ among clients.

## 3.1 Identifiability result

**Condition 3.4.** (Causal Minimality) Given the joint distribution $P(X)$, $P(X)$ is Markovian to a DAG $\mathcal{G}$ but not Markovian to any sub-graph of $\mathcal{G}$.

**Assumption 3.5.** Let a distribution $P(X)$ with $X = (X_1, X_2, \cdots, X_d)$ be induced from a restricted ANM with graph $\mathcal{G}$, and $P(X)$ satisfies *causal minimality* w.r.t $\mathcal{G}$.

**Assumption 3.6.** Let $P^{\mathcal{C}}(X)$ satisfy Assumption 3.1. At least one distribution $P^{c_k}(X) \in P^{\mathcal{C}}(X)$ meets Assumption 3.5 and the other distributions are faithful to $\mathcal{G}$.

**Proposition 3.7.** *Given $P^{\mathcal{C}}(X)$ satisfying Assumption 3.6, and then, $\mathcal{G}$ can be identified up from $P^{\mathcal{C}}(X)$.*

The detailed descriptions and the proof of Proposition 3.7 can be found in Appendix A.

## 4 Methodology

To solve the aforementioned problem, we formulate a continuous score-based method named DAG-shared federated causal discovery (DS-FCD). Firstly, we define an objective function that guides all models from different clients to federally learn the underlying causal graph $\mathcal{G}$ (or adjacency matrix $\boldsymbol{B}$), and at the same time also to learn personalized causal mechanisms for each client. As shown in Figure 2, for each client $c_k$, the local model consists of a causal graph learning part and a causal mechanisms approximation part. The CGL part is parameterized by a matrix $\boldsymbol{U}^{c_k} \in \mathbb{R}^{d \times d}$, which would be exactly the same for all clients finally[5]. To make every entry of $\boldsymbol{U}^{c_k}$ to approximate the binary entry of adjacency matrix, a Gumbel-Sigmoid method (Jang et al., 2017; Ng et al., 2022b) represented as $g_\tau(\boldsymbol{U}^{c_k})$, is further leveraged to transform $\boldsymbol{U}^{c_k}$ to a differentiable approximation of the adjacency matrix. The causal mechanisms $f_1^{c_k}, f_2^{c_k}, \cdots, f_d^{c_k}$ are parameterized by $d$ sub-networks, each of which has $d$ inputs and one output. In the learning procedure, the CGL parts (specifically $\boldsymbol{U}^{c_k}$) of participating clients are shared with the server $s$. Then, the processed information is broadcast to each client for self-updating its own matrix. The details of our method are demonstrated in the following subsections.

---

[5]Please notice that CGL parts of different clients may not be the same during the training procedure. So we index them.

## 4.1 The overall learning objective

Now we present the overall learning objective of FCD as the following optimization problem:

$$\underset{\boldsymbol{\Phi}, \boldsymbol{U}}{\arg\max} \quad \sum_{k=1}^{m} \mathcal{S}^{c_k}(\mathcal{D}^{c_k}, \boldsymbol{\Phi}^{c_k}, \boldsymbol{U}) \tag{6}$$

$$\text{subject to} \quad g_\tau(\boldsymbol{U}) \in \mathbf{DAGs} \;\Leftrightarrow\; h(\boldsymbol{U}) = \text{Tr}[e^{(g_\tau(\boldsymbol{U}))}] - d = 0,$$

where $\boldsymbol{\Phi}^{c_k} := \{\boldsymbol{\Phi}_1^{c_k}, \boldsymbol{\Phi}_2^{c_k}, \cdots, \boldsymbol{\Phi}_d^{c_k}\}$ represents the CMA part of the model on $c_k$. $\mathcal{S}^{c_k}(\cdot)$ is the scoring function for evaluating the fitness of local model of client $c_k$ and observations $\mathcal{D}^{c_k}$. For score-based causal discovery, selecting a proper score function such as BIC score (Schwarz, 1978), generalized score function (Huang et al., 2018) or equivalently taking the likelihood of $P(X)$ with a penalty function on model parameters (Zheng et al., 2018; Ng et al., 2022b; Zheng et al., 2020; Lachapelle et al., 2020) can guarantee to identify up the underlying ground-truth causal graph $\mathcal{G}$ because $\mathcal{G}$ is supposed to have the maximal score over Eq. (6). Throughout all experiments in this paper, we assume the noise type are Gaussian with equal variance for each local distribution. And, the overall score function utilized in this paper is as follows,

$$\mathcal{S}^{c_k}(\mathcal{D}^{c_k}, \boldsymbol{\Phi}^{c_k}, \boldsymbol{U}^{c_k}) = -\frac{1}{2n_k} \sum_{i=1}^{n_k} \sum_{j}^{d} \|\mathcal{D}_{ij}^{c_k} - \boldsymbol{\Phi}_j^{c_k}(g_\tau(\boldsymbol{U}_{j,:}^{c_k}) \circ \mathcal{D}_i^{c_k})\|_2^2 - \lambda \|g_\tau(\boldsymbol{U})\|_1 \tag{7}$$

In our score function, we take the negative Least Squares loss and a sparsity term, which corresponds to the model complexity penalty in the BIC score (Schwarz, 1978).[6] However, the global minimum is hard to reach by using gradient descent method due to the non-convexity of $h(\boldsymbol{U})$. More details on discussions of the optimization results can be found in the Appendix. C.

In this paper, instead of directly taking the likelihood of $P(X)$, we leverage the well-known results on the density transformation to model the distribution of $P(\mathcal{E})$, i.e., maximizing the likelihood $P^{c_k}(\mathcal{E}|\mathcal{F}^{c_k}, \mathcal{G})$ for $\forall c_k \in \mathcal{C}$. According to Eq. (1), we have $\epsilon_i = X_i - f_i(\mathbf{PA}_i)$. That is to say, modelling $P(\mathcal{E})$ can be achieved by an auto-regressive model. To get $\epsilon_i$, the first step is to select the parental set $\mathbf{PA}_i$ for $X_i$. This can be realized by $\boldsymbol{B}[:, i] \circ X$, where $\circ$ means the element-wise product. In our paper, for client $c_k$, we predict the noise by $\epsilon_i = X_i - \boldsymbol{\Phi}_i(g_\tau(\boldsymbol{U})[:, i] \circ X)$, where $g_\tau(\boldsymbol{U})$ is to approximate $\boldsymbol{B}$ and $\boldsymbol{\Phi}_i(\cdot)$ is parameterized by a neural network to approximate $f_i$. The specific formulation of $\mathcal{S}^{c_k}$ would depend on the assumption of noise distributions.

## 4.2 DAG-shared learning

As suggested in NOTEARS (Zheng et al., 2018), the hard-constraint optimization problem in Eq. (6) can be addressed by an Augmented Lagrangian Method (ALM) to get an approximate solution. Similar to the penalty methods, ALM transforms a constrained optimization problem by a series of unconstrained sub-problems and adds a penalty term to the objective function. ALM also introduces a Lagrangian multiplier term to avoid ill-conditioning by preventing the coefficient of penalty term from going too large. To solve Eq. (6), the sub-problem can be written as

$$\underset{\boldsymbol{\Phi}, \boldsymbol{U}}{\arg\max} \sum_{k=1}^{m} \mathcal{S}^{c_k}(\mathcal{D}^{c_k}, \boldsymbol{\Phi}^{c_k}, g_\tau(\boldsymbol{U})) - \alpha_t h(\boldsymbol{U}) - \frac{\rho_t}{2} h(\boldsymbol{U})^2, \tag{8}$$

where $\alpha_t$ and $\rho_t$ are the Lagrangian multiplier and penalty parameter of the $t$-th sub-problem, respectively. These parameters are updated after the sub-problem is solved. Since neural networks are adopted to fit the causal mechanisms in our work, there is no closed-form solution for Eq. (8). Therefore, we solve it approximately via Adam (Kingma & Ba, 2015). The method is described in Algorithms 1 and 2. And in Algorithm 1, we share the same coefficients updating strategy as in (Ng et al., 2022b).

Each sub-problem as Eq. (8) is solved mainly by distributing the computation across all local clients. Since data is prevented from sharing among clients and the server, each client owns its personalized model, which

---

[6]The consistency of BIC score for learning graphs on ANMs is discussed in Appendix C.5.

---

**Algorithm 1** DAG-Shared Federated Causal Discovery

---

1: **Input:** $\mathcal{D}, \mathcal{C}$, Parameter-list $= \{\alpha_{init}, \rho_{init}, h_{tol}, it_{max}, \rho_{max}, \beta, \gamma, r \}$
2: **Output:** $\mathbb{E}g_\tau(\boldsymbol{U}_t), \boldsymbol{\Phi}_t$
3: #Parameter Initializing
4: $t \leftarrow 1, \alpha_t \leftarrow \alpha_{init}, \rho_t \leftarrow \rho_{init}$
5: **while** $t \le it_{max}$ and $h(\boldsymbol{U}_t) \ge h_{tol}$ and $\rho \le \rho_{max}$ **do**
6:    #Sub-problem Solving
7:    $\boldsymbol{U}_{t+1}, \boldsymbol{\Phi}_{t+1} \leftarrow \mathrm{SPS}(\mathcal{D}, \mathcal{C}, \alpha_t, \rho_t, it_{in}, it_{fl}, r)$
8:    #Coefficients Updating
9:    $\alpha_{t+1} \leftarrow \alpha_t + \rho_t \mathbb{E}[h(\boldsymbol{U}_{t+1})], \quad t \leftarrow t+1$
10:    **if** $\mathbb{E}[h(\boldsymbol{U}_{t+1})] > \gamma \mathbb{E}[h(\boldsymbol{U}_t)]$ **then**
11:      $\rho_{t+1} = \beta \rho_t$
12:    **else**
13:      $\rho_{t+1} = \rho_t$
14:    **end if**
15: **end while**

---

**Algorithm 2** Sub-Problem Solver (SPS) for DS-FCD

---

1: **Input:** $\mathcal{D}, \mathcal{C}$, Parameter-list $= \{\alpha_t, \rho_t, it_{in}, it_{fl}, r\}$
2: **Output:** $\boldsymbol{U}_{new}, \boldsymbol{\Phi}^{it_{in}}$
3:  Define $\mathrm{SP}^{c_k} = \mathcal{S}^{c_k} - \alpha_t h(\boldsymbol{U}^{c_k}) - \frac{\rho_t}{2}h(\boldsymbol{U}^{c_k})^2$
4: **for** $i$ in $(1, 2, \cdots, it_{in})$ **do**
5:    **for each** client $c_k$ **do**
6:      #Self-updating
7:      $\boldsymbol{U}^{i,c_k}, \boldsymbol{\Phi}^{i,c_k} \leftarrow \arg\max_{\boldsymbol{\Phi}^{c_k}, \boldsymbol{U}^{c_k}} \mathrm{SP}^{c_k}$
8:    **end for**
9:    **if** $i \ (\% \ it_{fl}) = 0$ or $i = it_{in}$ **then**
10:      #Aggregating: randomly select $r$ clients and collect their $\boldsymbol{U}$s into $\mathbb{U}$, then, send $\mathbb{U}$ to the server
11:      $\mathbb{U} \leftarrow \mathrm{Agg}(r, \mathcal{C})$
12:      #Server Updating: average $\boldsymbol{U} \in \mathbb{U}$
13:      $\boldsymbol{U}_{new} \leftarrow \mathrm{Avg}(\mathbb{U})$
14:      #Broadcasting: distribute $\boldsymbol{U}_{new}$ to all clients
15:      $\mathcal{C} \leftarrow \mathrm{BD}(\boldsymbol{U}_{new})$
16:      **for each** client $c_k$ **do**
17:        #Clients Updating
18:        $\boldsymbol{U}^{i,c_k} \leftarrow \boldsymbol{U}_{new}$
19:      **end for**
20:    **end if**
21: **end for**

---

is only trained on its personalized data. The server communicates with clients by exchanging the parameters information of models and coordinates the joint learning task. To achieve so, our method alternately updates the server and clients in each communication round.

**Client Update.** For each model of client $c_k$, there are two main parts, named CGL part parameterized by $\boldsymbol{U}^{c_k}$ and CMA part parameterized by $\boldsymbol{\Phi}^{c_k}$, respectively. Essentially, the joint objective in Eq. (8) guides the learning process. In the self-updating as described in Algorithm 2, the clients firstly receive the updated penalty coefficients $\alpha_t$ and $\rho_t$ and the averaged parameter $\boldsymbol{U}_{new}$. Then, the renewed learning personalized score of client $c_k$ is defined as $\mathrm{SP}^{c_k} = \mathcal{S}^{c_k} - \alpha_t h(\boldsymbol{U}^{c_k}) - \frac{\rho_t}{2}h(\boldsymbol{U}^{c_k})^2$. $it_{fl}$ times of local gradient-based parameter updates are operated to maximize its personalized score.

**Server Update.** After $it_{fl}$ local updates, the server randomly chooses $r$ clients to collect their $\boldsymbol{U}$s to the set $\mathbb{U}$. Then, $\boldsymbol{U}$s in $\mathbb{U}$ are averaged to get $\boldsymbol{U}_{new}$. The other operation on the server is updating the $\alpha_t, \rho_t$ to

$\alpha_{t+1}, \rho_{t+1}$. The detailed calculating rules are described at lines $8-14$ in Algorithm 1. Then, the new penalty coefficients and parameter are broadcast to all clients. Notice that with assuming that data distribution across clients is IID, $\mathbf{\Phi}^{c_k}$ of the chosen $r$ clients can also be collected and averaged to update the local models of client in the same way, which is named as All-Shared FCD (AS-FCD) in this paper. It is worth noting that AS-FCD can further enhance the performance in the IID case but introduce some additional communication costs.

### 4.3 Thresholding

For continuous optimization, as illustrated in the previous work (Ng et al., 2022b), we leverages Gumbel-Sigmoid to approximate the binary mask. That is to say, the exact 0 or 1 is hard to get. The other issue is raised by ALM since the solution of ALM just satisfies the numerical precision of the constraint. This is because we set $h_{tol}$ and $it_{max}$ maximally but not infinite coefficients of penalty terms to formulate the last sub-problem. Therefore, some entries of the output $\boldsymbol{M} = \mathbb{E}g_\tau(\boldsymbol{U})$ will be near but not exactly 0 or 1. To alleviate this issue, $\ell_1$ sparsity is added to the objective function. In our method, since all mask values are in $[0, 1]$, we just take the middle value 0.5 as the threshold to prune the edges, which follows the same way in our baseline method MCSL (Ng et al., 2022b). The iterative thresholding method is also taken to deal with the case that the learned graph is cyclic. This may happen if the number of variables is large (40 variables in our paper). Because, in the numerical optimization, the constraint penalty exponentially decreases with the number of variables. To deal with the cyclic graph, we one-by-one cut edge with the minimum value until the graph is acyclic. To our knowledge, until now, all continuous search methods for causal discovery suffer from these two problems. It is an interesting future direction to be investigated.

### 4.4 Privacy and Costs Discussion

**Privacy issues of DS-FCD.** The strongest motivation of FL is to avoid *personalized raw data leakage*. To achieve this, DS-FCD proposes to exchange the parameters for modelling the causal graph. Here, we argue that the information leakage of local data is rather limited. The server, receiving parameters with client index, may infer some data property. However, according to the data generation model (1), the distribution of local data is decided by (1) causal graph, (2) noise types/strengths and (3) causal mechanisms. The gradient information of the shared matrix is decided by (1) the type of learning objective and (2) model architecture, which are agnostic to the server. Especially for the network part, clients may choose different networks to make the inference more complex. Moreover, if the causal graph is also private information for clients, this problem can be easily solved by selecting a client to serve as the proxy server. For the proxy server, it needs to play two roles, including training its own model and taking the server's duties. Then, in the communication round, other clients communicate with the proxy server instead of a real server. Moreover, the aim of our work, and federated learning in general, is not to provide a full solution to privacy protection. Instead, it is a first step towards this goal, i.e., no sharing of data between clients. To further protect privacy, more constraints need to be added to the federated learning framework, such as the prevention of information leakage from gradient sharing, which are studied under the privacy umbrella. To further enhance privacy protection, our method can also include more advanced privacy protection techniques (Wei et al., 2020b), which would be an interesting work to be investigated.

**Communication cost.** Since DS-FCD requires exchanging parameters between the server and clients, additional communication costs are raised. In our method, however, we argue that DS-FCD only brings rather small additional communication pressures. For the case of $d$ variables, a single communication only exchanges a $d \times d$ matrix twice (sending and receiving). For the IID setting, which assumes that local data are sampled from the same distribution, one can also transmit the neural network together to further improve the performance since causal mechanisms are also shared among clients. The trade-off between performance and communication costs can also be controlled by $r$ in Algorithm 2, i.e., enlarging or reducing $r$. Surprisingly, we find that reducing $r$ does not harm the performance severely (see Table 16 in Appendix D for detailed results). Moreover, the partial communication method, which only chooses some clients to exchange training information, is also leveraged to address the issue that not all clients are always online at the same time.

# 5 Experimental Results

In this section, we study the empirical performances of DS-FCD on both synthetic and real-world data. More detailed ablation experiments also can be found in Appendix D.

**Baselines** We compare our method with various baselines including some continuous search methods, named NOTEARS (Zheng et al., 2018), NOTEARS-MLP (N-S-MLP, for short) (Zheng et al., 2020), DAG-GNN (Yu et al., 2019) and MCSL (Ng et al., 2022b), and also two traditional combinatorial search methods named PC (Spirtes et al., 2001) and GES (Chickering, 2002). The comparison results with another method named causal additive models (CAM) (Bühlmann et al., 2014) are put in Appendix D.6. Furthermore, we also include a concurrent work named NOTEARS-ADMM (Ng & Zhang, 2022), which also considers learning Bayesian network in the federated setup. Since NOTEARS-ADMM focus more on the IID and linear settings and pays less attention to the nonlinear cases, we only include the results on linear cases of NOTEARS-ADMM in the main paper for fair comparisons. More detailed comparisons are shown in Appendix D.7. Moreover, we also compare our FCD with a voting method (Na & Yang, 2010) in Appendix D.5, which also tries to learn DAG from decentralized data. We provide two training ways for these compared methods. The first way is using all data to train only one model, which, however, is not permitted in FCD since the ban of data sharing in our setting. For the IID data, the results on this setting can be an *approximate upper bound* of our method but unobtainable. The second one is separately training each local model over its personalized data, of which the performances reported are the average results of all clients.

**Metrics.** We report two metrics named Structural Hamming Distance (SHD) and True Positive Rate (TPR) averaged over 10 random repetitions to evaluate the discrepancies between estimated DAG and the ground-truth graph $\mathcal{G}$. See more details about SHD, TPR in Appendix B.2 Notice that PC and GES can only reach the completed partially DAG (CPDAG, or MEC) at most, which shares the same Skelton with the ground-truth DAG $\mathcal{G}$. When we evaluate SHD, we just ignore the direction of undirected edges learned by PC and GES. That is to say, these two methods can get SHD 0 if they can identify the CPDAG. The implementation details of all methods are detailed in Appendix B.

## 5.1 Synthetic data

The synthetic data we consider here is generated from Gaussian ANMs (Model (1)). Two random graph models named Erdős-Rényi (ER) and Scale-Free (SF) (detailed definitions are shown in Appendix B.1.) are adopted to generate causal graph $\mathcal{G}$. And then, for each node $V_i$ corresponding to $X_i$ in $\mathcal{G}$, we sample a function from the given function sets to simulate $f_i$. Finally, data are generated according to a specific sampling method. In the following experiments, we take 10 clients and each with 600 observations (unless otherwise specified in some ablation studies.) throughout this paper. According to Assumption 3.1, data across all clients share the same causal graph for both IID and heterogeneous data settings. Due to the space limit, more ablation experiments, such as *uneven distributed observations*, *varying clients*, *dense graph*, *different non-linear functions*, and *different number of observations*, etc., are put in Appendix D. All detailed discussions on the experimental results are in Appendix E.

### 5.1.1 IID setting

**Results on linear model.** For fair comparison, here, we also provide the linear version of our method. Since linear data are parameterized with an adjacency matrix, we can directly take the adjacency matrix as our model instead of a CGL part and a CMA part. During training, the matrix are communicated and averaged by the server to coordinate the joint learning procedures.

NOTEARS-ADMM is also a causal discovery method designed for learning Bayesian network from decentralized data. Different from our average strategy to exchange training information among clients, ADMM is used for address this issue. From Table 1, we find that our method can consistently show its advantage on the linear case. In the *ER2 with 10 nodes* setting, our AS-FCD is even better than NOTEARS with all training data. While it is possible and the detailed explanation can be found in Appendix E.

Table 1: Results on the linear model (IID).

| | ER2 with 10 nodes | | SF2 with 10 nodes | | ER2 with 20 nodes | | SF2 with 20 nodes | |
|---|---|---|---|---|---|---|---|---|
| | SHD ↓ | TPR ↑ | SHD ↓ | TPR ↑ | SHD ↓ | TPR ↑ | SHD ↓ | TPR ↑ |
| NOTEARS-All | $1.6 \pm 1.6$ | $0.93 \pm 0.06$ | *1.4 ± 1.1* | *0.92 ± 0.05* | *3.0 ± 2.7* | *0.94 ± 0.06* | *6.9 ± 7.0* | *0.86 ± 0.12* |
| NOTEARS-Sep | $3.0 \pm 2.2$ | $0.85 \pm 0.08$ | $3.6 \pm 2.1$ | $0.83 \pm 0.10$ | $4.1 \pm 2.4$ | $0.91 \pm 0.05$ | $10.2 \pm 5.9$ | $0.82 \pm 0.10$ |
| NOTEARS-ADMM | $4.7 \pm 3.9$ | $0.89 \pm 0.12$ | $4.4 \pm 3.0$ | $0.86 \pm 0.09$ | $7.9 \pm 5.9$ | $0.89 \pm 0.07$ | $10.7 \pm 5.3$ | $0.82 \pm 0.08$ |
| AS-FCD | $\mathbf{1.3 \pm 1.5}$ | $\mathbf{0.94 \pm 0.07}$ | $\mathbf{1.6 \pm 0.97}$ | $\mathbf{0.91 \pm 0.06}$ | $\mathbf{3.9 \pm 3.1}$ | $0.91 \pm 0.06$ | $\mathbf{9.4 \pm 6.7}$ | $0.82 \pm 0.12$ |

Table 2: Results on nonlinear ANM with GP (IID).

| | | ER2 with 10 nodes | | SF2 with 10 nodes | | ER2 with 40 nodes | | SF2 with 40 nodes | |
|---|---|---|---|---|---|---|---|---|---|
| | | SHD ↓ | TPR ↑ | SHD ↓ | TPR ↑ | SHD ↓ | TPR ↑ | SHD ↓ | TPR ↑ |
| All data | PC | $15.3 \pm 2.6$ | $0.37 \pm 0.10$ | $14.1 \pm 4.3$ | $0.44 \pm 0.20$ | $84.9 \pm 13.4$ | $0.40 \pm 0.08$ | $95.0 \pm 10.4$ | $0.36 \pm 0.07$ |
| | GES | $13.0 \pm 3.9$ | $0.50 \pm 0.18$ | $9.6 \pm 4.4$ | $0.71 \pm 0.17$ | $59.0 \pm 9.8$ | $0.53 \pm 0.08$ | $73.8 \pm 11.9$ | $0.47 \pm 0.10$ |
| | NOTEARS | $16.5 \pm 2.0$ | $0.05 \pm 0.04$ | $14.5 \pm 1.1$ | $0.09 \pm 0.07$ | $71.2 \pm 7.2$ | $0.08 \pm 0.03$ | $70.8 \pm 2.3$ | $0.07 \pm 0.03$ |
| | N-S-MLP | $8.1 \pm 3.8$ | $0.56 \pm 0.17$ | $8.3 \pm 2.8$ | $0.51 \pm 0.16$ | $45.3 \pm 6.8$ | $0.43 \pm 0.08$ | $49.2 \pm 7.7$ | $0.39 \pm 0.09$ |
| | DAG-GNN | $16.2 \pm 2.1$ | $0.07 \pm 0.06$ | $15.2 \pm 0.8$ | $0.05 \pm 0.05$ | $73.0 \pm 7.7$ | $0.06 \pm 0.03$ | $72.4 \pm 1.6$ | $0.05 \pm 0.02$ |
| | MCSL | $1.9 \pm 1.5$ | *0.90 ± 0.08* | *1.6 ± 1.2* | *0.91 ± 0.07* | *25.4 ± 13.1* | *0.68 ± 0.14* | $31.6 \pm 10.0$ | *0.59 ± 0.13* |
| Sep data | PC | $14.1 \pm 2.4$ | $0.31 \pm 0.06$ | $13.6 \pm 2.7$ | $0.30 \pm 0.10$ | $83.8 \pm 7.4$ | $0.24 \pm 0.03$ | $86.1 \pm 4.6$ | $0.23 \pm 0.04$ |
| | GES | $12.7 \pm 2.7$ | $0.37 \pm 0.09$ | $12.7 \pm 2.4$ | $0.33 \pm 0.11$ | $71.0 \pm 6.7$ | $0.29 \pm 0.03$ | $73.2 \pm 4.4$ | $0.29 \pm 0.05$ |
| | NOTEARS | $16.5 \pm 2.0$ | $0.06 \pm 0.04$ | $14.6 \pm 1.0$ | $0.09 \pm 0.06$ | $71.1 \pm 7.3$ | $0.08 \pm 0.03$ | $70.7 \pm 2.0$ | $0.07 \pm 0.03$ |
| | N-S-MLP | $8.5 \pm 2.9$ | $0.56 \pm 0.13$ | $8.7 \pm 2.9$ | $0.53 \pm 0.16$ | $51.0 \pm 6.9$ | $0.41 \pm 0.06$ | $53.6 \pm 5.5$ | $0.39 \pm 0.08$ |
| | DAG-GNN | $15.7 \pm 2.3$ | $0.11 \pm 0.05$ | $14.5 \pm 1.0$ | $0.10 \pm 0.06$ | $71.5 \pm 7.5$ | $0.08 \pm 0.02$ | $70.8 \pm 1.8$ | $0.07 \pm 0.02$ |
| | MCSL | $7.1 \pm 3.2$ | $0.83 \pm 0.08$ | $6.9 \pm 2.8$ | $0.84 \pm 0.08$ | $77.3 \pm 19.8$ | $0.64 \pm 0.11$ | $72.9 \pm 16.4$ | $\mathbf{0.58 \pm 0.13}$ |
| | DS-FCD | $2.4 \pm 2.0$ | $0.86 \pm 0.13$ | $2.7 \pm 2.2$ | $0.86 \pm 0.13$ | $36.5 \pm 12.1$ | $0.65 \pm 0.15$ | $46.4 \pm 10.4$ | $0.57 \pm 0.13$ |
| | AS-FCD | $\mathbf{1.8 \pm 2.0}$ | $\mathbf{0.89 \pm 0.12}$ | $\mathbf{2.5 \pm 2.7}$ | $\mathbf{0.85 \pm 0.15}$ | $\mathbf{30.0 \pm 12.3}$ | $\mathbf{0.74 \pm 0.15}$ | $\mathbf{31.5 \pm 10.0}$ | $\mathbf{0.59 \pm 0.13}$ |

**Results on nonlinear model.** For the IID setting, all data are generated by an ANM and divided into 10 pieces. Each $f_i$ is sampled from a Gaussian Process (GP) with RBF kernel of bandwidth one (See Table 13 and Table 14 in Appendix. D for results of other functions.) and noises are sampled from one zero-mean Gaussian distribution with fixed variance. We consider graphs of $d$ nodes and $2d$ expected edges.

Experimental results are reported in Table 2 with nodes 10 and 40. Since all local data are IID, here, we also provide another effective training method named AS-FCD, in which the CMA parts are also shared among clients. In all settings, AS-FCD shows a better performance than DS-FCD due to that more model information are shared during training. While DS-FCD can also show a consistent advantage over other methods. When separately training local models, all models suffer from data scarcity. Therefore, we can observe that both DS-FCD and AS-FCD perform better than other methods in the fashion of separate training. NOTEARS and DAG-GNN, as continuous search methods, obtain unsatisfactory results due to the weak model capacity and improper model assumption. While BIC score of GES gets a linear-Gaussian likelihood, which is incapable to deal with non-linear data[7]. With the number of nodes increasing, DS-FCD still shows better results than the closely-related baseline method MCSL. However, NOTEAES-MLP can show a comparable result with DS-FCD owing to the advantage over MCSL.

### 5.1.2 **Heterogeneous data** setting

As defined in Section 3, the heterogeneous data property of data across clients come from the changes of causal mechanisms or the shift of noise distributions. To simulate the heterogeneous data, we firstly generate a DAG shared by all clients and then decide the types of causal mechanisms $f_i^{c_k}$ and noises $\epsilon_i$ for $i \in [d]$ for each client $c_k$. In our experiments, We allow that $f^{c_k}$ can be linear or non-linear for each client. If being linear, $f^{c_k}$ here is a weighted adjacency matrix with coefficients sampled from Uniform $([-2.0, -0.5] \cup [0.5, 2.0])$, with equal probability. If being non-linear, $f_i^{c_k}$ is independently sampled from GP, GP-add, MLP or MIM functions (Yuan, 2011), randomly. Then, a fixed zero-mean Gaussian noise is set to each client with a randomly sampled variance from $\{0.8, 1\}$.

---

[7]Please find the ablation experiment with linear data and more discussions of the experimental results in Appendix D.

Table 3: Results on ANMs with heterogeneous data.

|  |  | ER2 with 10 nodes | | SF2 with 10 nodes | | ER2 with 40 nodes | | SF2 with 40 nodes | |
|---|---|---|---|---|---|---|---|---|---|
|  |  | SHD ↓ | TPR ↑ | SHD ↓ | TPR ↑ | SHD ↓ | TPR ↑ | SHD ↓ | TPR ↑ |
| All data | PC | $22.3 \pm 4.2$ | $0.41 \pm 0.11$ | $21.0 \pm 3.6$ | $0.41 \pm 0.12$ | $151.9 \pm 14.2$ | $0.27 \pm 0.08$ | $152.5 \pm 5.4$ | $0.26 \pm 0.04$ |
|  | GES | $26.4 \pm 6.2$ | $0.53 \pm 0.14$ | $25.4 \pm 4.6$ | $0.54 \pm 0.13$ | NaN | NaN | NaN | NaN |
|  | NOTEARS | $20.4 \pm 4.1$ | $0.49 \pm 0.14$ | $18.7 \pm 3.3$ | $0.45 \pm 0.11$ | $164.8 \pm 47.4$ | $0.39 \pm 0.07$ | $178.1 \pm 33.0$ | $0.40 \pm 0.10$ |
|  | N-S-MLP | $22.8 \pm 5.0$ | $0.87 \pm 0.07$ | $24.7 \pm 3.3$ | $0.88 \pm 0.07$ | $344.4 \pm 71.9$ | *$0.92 \pm 0.08$* | $325.0 \pm 50.2$ | *$0.85 \pm 0.08$* |
|  | DAG-GNN | $21.2 \pm 6.0$ | $0.39 \pm 0.11$ | $16.6 \pm 3.0$ | $0.48 \pm 0.18$ | $146.6 \pm 41.6$ | $0.29 \pm 0.08$ | $168.2 \pm 34.2$ | $0.31 \pm 0.09$ |
|  | MCSL | $19.4 \pm 4.4$ | $0.75 \pm 0.19$ | $19.0 \pm 4.0$ | $0.81 \pm 0.14$ | $118.6 \pm 18.1$ | $0.68 \pm 0.11$ | $126.9 \pm 16.5$ | $0.59 \pm 0.12$ |
| Sep data | PC | $12.5 \pm 2.7$ | $0.45 \pm 0.07$ | $11.0 \pm 2.1$ | $0.49 \pm 0.07$ | $65.7 \pm 11.0$ | $0.43 \pm 0.06$ | $73.7 \pm 5.5$ | $0.36 \pm 0.05$ |
|  | GES | $12.9 \pm 2.6$ | $0.58 \pm 0.07$ | $10.3 \pm 2.8$ | $0.60 \pm 0.09$ | $68.2 \pm 20.8$ | $0.65 \pm 0.09$ | $77.2 \pm 13.8$ | $0.60 \pm 0.07$ |
|  | NOTEARS | $7.6 \pm 2.6$ | $0.60 \pm 0.11$ | $7.6 \pm 1.8$ | $0.58 \pm 0.09$ | $\mathbf{34.9 \pm 12.7}$ | $0.63 \pm 0.11$ | $\mathbf{43.4 \pm 8.4}$ | $0.53 \pm 0.10$ |
|  | N-S-MLP | $\mathbf{5.2 \pm 1.4}$ | $\mathbf{0.80 \pm 0.05}$ | $\mathbf{6.1 \pm 1.6}$ | $\mathbf{0.76 \pm 0.05}$ | $46.0 \pm 10.2$ | $\mathbf{0.73 \pm 0.08}$ | $56.0 \pm 9.5$ | $\mathbf{0.66 \pm 0.09}$ |
|  | DAG-GNN | $8.2 \pm 2.9$ | $0.67 \pm 0.12$ | $8.4 \pm 2.1$ | $0.67 \pm 0.09$ | $45.7 \pm 13.5$ | $0.64 \pm 0.11$ | $52.7 \pm 8.4$ | $0.60 \pm 0.11$ |
|  | MCSL | $9.2 \pm 1.8$ | $0.72 \pm 0.06$ | $8.9 \pm 2.0$ | $0.71 \pm 0.08$ | $76.1 \pm 13.7$ | $0.53 \pm 0.09$ | $78.1 \pm 6.3$ | $0.47 \pm 0.07$ |
|  | DS-FCD | $\mathbf{1.9 \pm 1.6}$ | $\mathbf{0.99 \pm 0.02}$ | $\mathbf{2.6 \pm 1.3}$ | $\mathbf{0.93 \pm 0.07}$ | $\mathbf{24.3 \pm 10.2}$ | $\mathbf{0.86 \pm 0.09}$ | $\mathbf{33.9 \pm 10.9}$ | $\mathbf{0.73 \pm 0.09}$ |

We can see that the conclusion of experimental results on the heterogeneous data setting is rather similar to that of the IID. As can be read from Table 3, DS-FCD always shows the best performances across all settings. If taking all data together to train one model using other methods, we can see that data heterogeneity would put great trouble to all compared methods while DS-FCD plays pretty well. Moreover, DS-FCD shows consistent good results with different numbers of observations on each client (see Table 15). NOTEARS takes second place at the setting of 40 nodes because there are some linear data among clients, which is also the reason that DS-FCD shows lower SHDs on heterogeneous data in Table 3 than Table 2. Compared with Non-linear models, NOTEARS easily fits well with even fewer linear data.

## 5.2 Real data

We consider a real public dataset named **fMRI Hippocampus** (Poldrack et al., 2015) to discover the causal relations among six brain regions. This dataset records signals from six separate brain regions in the resting state of one person in 84 successive days and the anatomical structure provides 7 edges as the ground truth graph (see Figure 10 in Appendix D). Herein, we separately select 500 records in each of 10 days (see Figure 12 for the normalized data distribution in Appendix D), which can be regarded as different local data. It is worth noting that though this data does not have a real data privacy problem, we can use this dataset to evaluate the learning accuracy of our method. Here, in Table 4 we show part of the experimental results while others lie in Table 17 (Appendix D). AS-FCD shows the best performance over all criterion while DS-FCD also performs better than most of the other methods.

Table 4: Empirical results on **fMRI Hippocampus** dataset (Part 1).

|  | All data | | | Separate data | | | DS-FCD | AS-FCD |
|---|---|---|---|---|---|---|---|---|
|  | PC | NOTEARS | MCSL | PC | NOTEARS | MCSL |  |  |
| SHD ↓ | $9.0 \pm 0.0$ | *$5.0 \pm 0.0$* | $9.0 \pm 0.6$ | $8.7 \pm 1.3$ | $8.0 \pm 1.9$ | $8.3 \pm 1.7$ | $\mathbf{6.4 \pm 0.9}$ | $\mathbf{5.0 \pm 0.0}$ |
| NNZ | $11.0 \pm 0.0$ | $4.0 \pm 0.0$ | $12.0 \pm 0.6$ | $7.6 \pm 1.3$ | $5.4 \pm 1.5$ | $9.0 \pm 1.7$ | $6.8 \pm 0.6$ | $5.0 \pm 0.0$ |
| TPR ↑ | $0.43 \pm 0.00$ | $0.29 \pm 0.00$ | *$0.44 \pm 0.04$* | $0.26 \pm 0.11$ | $0.19 \pm 0.18$ | $\mathbf{0.35 \pm 0.15}$ | $0.27 \pm 0.12$ | $\mathbf{0.29 \pm 0.00}$ |
| FDR ↓ | $0.73 \pm 0.00$ | *$0.50 \pm 0.00$* | $0.74 \pm 0.03$ | $0.76 \pm 0.10$ | $0.78 \pm 0.19$ | $0.73 \pm 0.11$ | $\mathbf{0.72 \pm 0.11}$ | $\mathbf{0.60 \pm 0.00}$ |

## 6 Related work

Two mainstreams named constraint-based and score-based methods push the development of causal discovery. Constraint-based methods, including PC and fast causal inference (FCI) (Spirtes et al., 2001), take conditional independence constraints induced from the observed distribution to decide the graph skeleton and part of the directions. Another branch of methods (Chickering, 2002) define a score function, which evaluate the fitness between the distribution and graph, and identify the graph $\mathcal{G}$ with the highest score after searching

the DAG space. To avoid solving the combinatorial optimization problem, NOTEARS (Zheng et al., 2018) introduces an equivalent acyclicity constraint and formulates a fully continuous optimization for searching the graph. Following this work, many works leverages this constraint to non-linear case (Ng et al., 2019; Zheng et al., 2020; Lachapelle et al., 2020; Zhu et al., 2020; Wang et al., 2021; Gao et al., 2021; Ng et al., 2022b), low-rank graph (Fang et al., 2020), interventional data (Brouillard et al., 2020; Ke et al., 2019; Scherrer et al., 2021), time-series data (Pamfil et al., 2020), incomplete data (Gao et al., 2022; Geffner et al., 2022) and unmeasured confounding (Bhattacharya et al., 2021). GOLEM (Ng et al., 2020) leverages the full likelihood and soft constraint to solve the optimization problem. Ng et al. (2022a), DAG-NoCurl (Yu et al., 2021) and NOFEARS (Wei et al., 2020a) focus on the optimization aspect.

The second line of related work is on the Overlapping Datasets (OD) (Danks et al., 2009; Tillman & Spirtes, 2011; Triantafillou & Tsamardinos, 2015; Huang et al., 2020a) problem in causal discovery. However, OD assumes that each dataset owns observations of partial variables and targets learning the integrated DAG from multiple datasets. In these works, data from different sites need to be collected on a central server.

The last line is on federated learning (Yang et al., 2019; Kairouz et al., 2021), which provides the joint training paradigm to learn from decentralized data while avoiding sharing raw data during the learning process. FedAvg (McMahan et al., 2017) first formulates and names federated learning. FedProx (Li et al., 2020) studies the Non-IID case and provides the convergence analysis results. SCAFFOLD leverages variance reduction by correcting client-shift to enhance the training efficiency. Besides these fundamental problems in FL itself, this novel learning way has been widely co-operated with or applied to many real-world tasks such as healthcare (Sheller et al., 2020), recommendation system (Yang et al., 2020), and smart transport (Samarakoon et al., 2019).

### 6.1 Concurrent work (NOTEARS-ADMM)

In NOTEARS-ADMM (Ng & Zhang, 2022), the authors also consider the totally same setting that how to discover the causal relations from distributed data owing to privacy and security concerns.

The main advantage of our FCD over NOTEARS-ADMM is to handle with heterogeneous data, which is very common in real applications. Then, NOTEARS-ADMM mainly consider the linear case, which actually share the same learning object with our method. Instead of taking average to share training information, ADMM is taken to make the adjacency matrix close. More detailed experimental comparisons can be found in Appendix D.7, from which we can see that our FCD shows better performances in most of settings.

## 7 Conclusion and Discussions

Learning causal structures from decentralized data brings huge challenges to traditional causal discovery methods. In this context, we have introduced the first federated causal discovery method called DS-FCD, which uses a two-level structure for each local model. During the learning procedure, each client tries to learn an adjacency matrix to approximate the causal graph and neural networks to approximate the causal mechanisms. The matrix parts of some participating clients are aggregated and processed by the server and then broadcast to each client for updating its personalized matrix. The overall problem is formulated as a continuous optimization problem and solved by gradient descent methods. Structural identifiability conditions are provided and extensive experiments on various data sets to show the effectiveness of our DS-FCD.

The first limitation of our framework is with the causal sufficiency assumption, which is seldom right in real scenarios. While, as a general framework, the advanced methods (Bhattacharya et al., 2021), which can handle the no observed confounder case, can be well incorporated with our method to deal with the federated setup (More details can be seen in Appendix C.4). In federated learning, FedAvg with heterogeneous data (Liang et al., 2020) are well analyzed. However, the non-convexity property of our FCD comes both from the DAG constraint and nonlinear causal mechanisms. Therefore, the other interesting future work is to investigate the convergence properties of our proposed method.

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

# Appendix

# Table of Contents

# A    Structure identifiability

Besides exploring effective causal discovery methods, identifiability conditions of causal models (Spirtes et al., 2001) are also important. In general, unique identification of the ground truth DAG is impossible from purely observational data without some specific assumptions. However, accompanying some specific data generation assumptions, the causal graph can be identified (Peters et al., 2011; Peters & Bühlmann, 2014; Zhang & Hyvarinen, 2009; Shimizu et al., 2006; Hoyer et al., 2008). We first give the definition of identifiability in the decentralized setting.

**Definition A.1.** Consider a decentralized distribution set $P^{\mathcal{C}}(X)$ satisfying Assumption 3.1. Then, $\mathcal{G}$ is said to be identifiable if $P^{\mathcal{C}}(X)$ cannot be induced from any other DAG.

**Condition A.2.** (Cond. 19 in (Peters & Bühlmann, 2014)) The triple $(f_j, P(X_i), P(\epsilon_j))$ does not solve the following differential equation for all $x_i$, $x_j$ with $v''(x_j - f(x_i))f'(x_i) \neq 0$:

$$\xi''' = \xi'' \left( -\frac{\nu''' f'}{\nu''} + \frac{f''}{f'} \right) - 2\nu'' f'' f' + \nu' f''' + \frac{\nu' \nu''' f'' f'}{\nu''} - \frac{\nu' (f'')^2}{f'}.$$

Here, $f := f_j$ and $\xi := \log P(X_i)$, and $v := \log P(\epsilon_j)$ are the logarithms of the strictly positive densities.

**Definition A.3.** (Restricted ANM. Def. 27 in (Peters & Bühlmann, 2014)) Consider an ANM with $d$ variables. This SEM is called restricted ANM if for all $j \in \mathbb{V}, i \in \mathbf{PA}_j$ and all sets $\mathbb{S} \subseteq \mathbb{V}$ with $\mathbf{PA}_j \backslash \{i\} \subseteq \mathbb{S} \subseteq \mathbf{PA}_j \backslash \{i, j\}$, there is an $x_{\mathbb{S}}$ with $P(x_{\mathbb{S}}) > 0$, s.t. the tripe

$$\left( f_j(x_{\mathbf{PA}_j \backslash \{i\}}, \underbrace{\cdot}_{X_i}), P\left(X_i \mid X_{\mathbb{S}} = x_{\mathbb{S}}\right), P\left(\epsilon_j\right) \right)$$

satisfies ConditionA.2. Here, the under-brace indicates the input component of $f_j$ for variable $X_i$. In particular, we require the noise variables to have non-vanishing densities and the functions $f_j$ to be continuous and three times continuously differentiable.

## A.1    Proof of Proposition 3.7

From Remark 3.2, we have $P^{c_k}(X) \in P^{\mathcal{C}}(X)$ for $\forall c_k$, is Markov with $\mathcal{G}$. For each $c_k \in \mathcal{C}$ with $P^{c_k}(X)$ does not satisfy Assumption 3.5, the Completed Partially DAG (CPDAG) $\hat{\mathcal{G}}$ (Pearl, 2009), which represents the CPDAG induced by $\mathcal{G}$, can be identified (Spirtes et al., 2001). (1) That also says that these distributions can be induced from any DAG induced from $\mathcal{M}(\mathcal{G})$, including $\mathcal{G}$ definitely. Notice that skeleton($\hat{\mathcal{G}}$) = Skeleton($\mathcal{G}$) and any $X_i \leftarrow X_j$ in $\hat{\mathcal{G}}$ is also existed in $\mathcal{G}$. Then, for those $c_k$ with with $P^{c_k}(X)$ satisfying Assumption 3.5, $\mathcal{G}$ can be identified. (2) That is to say, distributions satisfying Assumption 3.5 can only be induced from $\mathcal{G}$. Then, two kinds of graph, $\hat{\mathcal{G}}$ and $\mathcal{G}$, are obtained. Therefore, $\mathcal{G}$ can be easily identified. With (1) and (2), $P^{c_k}(X) \in P^{\mathcal{C}}(X)$ for $\forall c_k$ can only be induced by $\mathcal{G}$. Then, $\mathcal{G}$ is said to be identifiable    ■

# B    Implementations

The comparing causal discovery methods used in this paper all have available implementations, listed below:

- PC and MCSL: Codes are available at `gCastle` https://github.com/huawei-noah/trustworthyAI/tree/master/gcastle. The first author of MCSL added the implementation in this package.

- NOTEARS and NOTEARS-MLP: Codes are available at the first author's GitHub repository https://github.com/xunzheng/notears

- NOTEARS-ADMM: Codes are available at the first author's GitHub repository https://github.com/ignavierng/notears-admm

- DAG-GNN: Codes are available at the author's GitHub repository https://github.com/fishmoon1234/DAG-GNN

- GES: an implementation of GES is available at https://github.com/juangamella/ges

- CAM: the codes are available at `CRAN R` package repository https://cran.r-project.org/src/contrib/Archive/CAM/

Our implementation is highly based on the existing Tool-chain named `gCastle` (Zhang et al., 2021), which includes many gradient-based causal discovery methods.

## B.1 Graph generation

To simulate DAG for generating observations, we introduce two kinds of graph generation methods named Erdős-Rényi (ER) and Scale-Free (SF) graphs. To simulate the ER graph generation, we firstly randomly sample a topological order and by adding directed edges were it is allowed independently with probability $p = \frac{2s}{d^2-d}$ where $s$ is the number of edges in the resulting DAG. To generate Scale-free (SF) graphs, we firstly take the Barabasi-Albert model and then add all nodes one by one. From the above descriptions, we can find that the degree distribution of ER graphs follows a Poisson distribution, and the degree of SF graphs follows a power law: few nodes, often called hubs, have a high degree (Lachapelle et al., 2020).

## B.2 Detailed metrics

SHD is kind of measurement which is defined to calculate the Hamming distance two partially directed acyclic graphs (PDAG) by counting the number of edge for which the edge type differs in both PDAGs. In PDAG, there exist four kinds of edges between two nodes: $i \to j$, $i \leftarrow j$, $i - j$ and $i\ j$. SHD just counts the different edges between the two graphs. SHD is defined over the space of PDAGs, so we can, of course, use it to calculate distances in DAG and CPDAG spaces.

True Positive Rate (TPR) and False Discovery Rate (FDR) are two common metrics in the machine learning community. True positive rate, also referred to sensitivity or recall, is used to measure the percentage of actual positives which are correctly identified. The False Discovery Rate is defined as the expected proportion of errors committed by falsely rejecting the null hypothesis. Let $TP$ be true positives (samples correctly classified as positive), and $FN$ be false negatives (samples incorrectly classified as negative), $FP$ be false positives (samples incorrectly classified as positive), and $TN$ be true negatives (samples correctly classified as negative). Then, $TPR = \frac{TP}{TP+FN}$ and $FDR = \frac{FP}{FP+TP}$.

## B.3 Hyper-parameters setting

In all experiments, there is no extra hyper-parameter to adjust for PC (with Fisher-z test and $p$-value 0.01) and GES (BIC score). For NOTEARS, NOTEARS-MLP and DAG-GNN, we use the default hyper-parameters provided in their papers/codes. For MCSL, the hyper-parameters need to be modified are $\rho_{init}$ and $\beta$. Specifically, if experimental settings (10 variables and 20 variables) are the same as those in their paper, we just take all the recommended hyper-parameters. For settings not implemented in their paper (40 variables exactly), we have two kinds of implementations. The first one is taking a linear interpolation for choosing the hyper-parameters. The second one is taking the same parameters as ours. We find that the second choice always works better. In our experiment, we report the experimental results done in the second way. Notice that CAM pruning is also introduced to improve the performance of MCSL, which however can not guarantee a better result in our settings. For simplicity and fairness, we just take the direct outputs of MCSL.

Similar to MCSL (Ng et al., 2022b) and GraN-DAG (Lachapelle et al., 2020), we implement several experiments on simulated data with known causal graphs to search for the hyper-parameters and then use these hyper-parameters for all the simulated experiments. Specifically, we use seeds from 1 to 10 to generate the simulated data to search for the best combination of hyperparameters while all our experimental results reported in this paper are all conducted using seeds from 2021 to 2030.

### B.4   Hyper-parameters in real-data setting

Most CSL methods have hyper-parameters, more or less, which need to be decided prior to learning. Moreover, NN-based methods are especially sensitive to the selection of hyper-parameters. For instance, Gran-DAG (Lachapelle et al., 2020) defines a really large hyper-parameters space for searching the optimal combination, which even uses different learning rates for the first subproblem and the other subproblems. MCSL and DS-FCD are sensitive to the selection of $\rho_{init}$ and $\beta$ when constructing and solving the subproblem. As pointed out in (Kairouz et al., 2021), NOTEARS focus more on optimizing the scoring term in the early stage and pays more attention to approximate DAG in the late stage. If NOTEARS cannot find a graph near $\mathcal{G}$ in the early stage, then, it would lead to a worse result.

To alleviate this problem, one may choose to (1) enlarge the learning rate or take more steps when solving the first few subproblems as Gran-DAG; (2) reduce the value of coefficient $\rho_{init}$ to let the optimizer pay more attention to the scoring term in the early stages as MCSL. The other trick we find when dealing with real data is increasing $\ell_1$. This mostly results from that real data may not fit well with the data generation assumptions in most papers. Therefore, we choose to conduct a grid search to find the best combination of $\rho_{init}, \beta, \ell_1$ for causal discovery on real data.

In the practice of causal discovery, it is impossible to have $\mathcal{G}$ to select the hyper-parameters. One common approach is trying multiple hyper-parameter combinations and keeping the one yielding the best score evaluated on a validation set (Koller & Friedman, 2009; Ng et al., 2022b; Lachapelle et al., 2020). However, the direct use of this method may not work for some algorithms, such as MCSL, NOTEARS-MLP, and DS-FCD. This mainly lies in the similar explanations of the property of the traditional solution of AL. In the late stage of optimization, the optimizer focuses heavily on finding *a DAG* by enlarging the penalty coefficient $\rho$. Then, the learning of causal mechanisms would be nearly ignored. To address this problem, we firstly report the DAG directly learned by a combination of hyper-parameters. And then, we replace the parameters part for describing the causal graph with the learned DAG. Afterwards, we just take the score without DAG constraint to optimize the causal mechanism approximation part (which may not be the same name in the other algorithms). Finally, the validation set is taken to evaluate the learned model. The final hyper-parameters used on the real dataset in our paper is as follows:

Table 5: The hyper-parameters used on real data.

| Parameters | $\rho_{init}$ | $\beta$ | $\lambda_{\ell_1}$ |
|---|---|---|---|
| Values | 0.008 | 2 | 0.3 |

### B.5   Model parameters

The CGL part in each local model is parameterized by a $d \times d$ matrix named $\boldsymbol{U}$ and the Gumbel-Sigmoid approach is leveraged for approximating the binary form. Each entry in $\boldsymbol{U}$ is initialized as 0. The temperature $\tau$ is set to 0.2 for all settings. Then, for the causal mechanism approximation part, we use 4 dense layers with 16 variables in each hidden layer. All weights in the Network are initialized using the Xavier uniform initialization.

### B.6   Training prameters

Our AS-FCD and DS-FCD reach this point and are implemented with the following hyper-parameters. We take Adam (Kingma & Ba, 2015) with learning rate $3 \times 10^{-2}$ and all the observational data $\mathcal{D}^{c_k}$ on each client are used for computing the gradient. And the detailed parameters used in Algorithms 1 and 2 are listed in Table 6.

Notice that as illustrated in MCSL (Ng et al., 2022b), the performance of the algorithm is affected by the initial value of $\rho_{init}$ and the choice of $\beta$. Since a small initial of $\rho_{init}$ and $\beta$ would result in a rather long training time. As said in (Kaiser & Sipos, 2021), MLE plays an important role in the early stage of training

Table 6: The hyper-parameters used on simulated data in this paper.

| Parameters | $\alpha_{init}$ | $h_{tol}$ | $it_{max}$ | $it_{inner}$ | $it_{fl}$ | $\gamma$ | $\rho_{max}$ | $\lambda_{\ell_1}$ |
|---|---|---|---|---|---|---|---|---|
| Values | 0 | $1 \times 10^{-10}$ | 25 | 1000 | 200 | 0.25 | $1 \times 10^{14}$ | 0.01 |

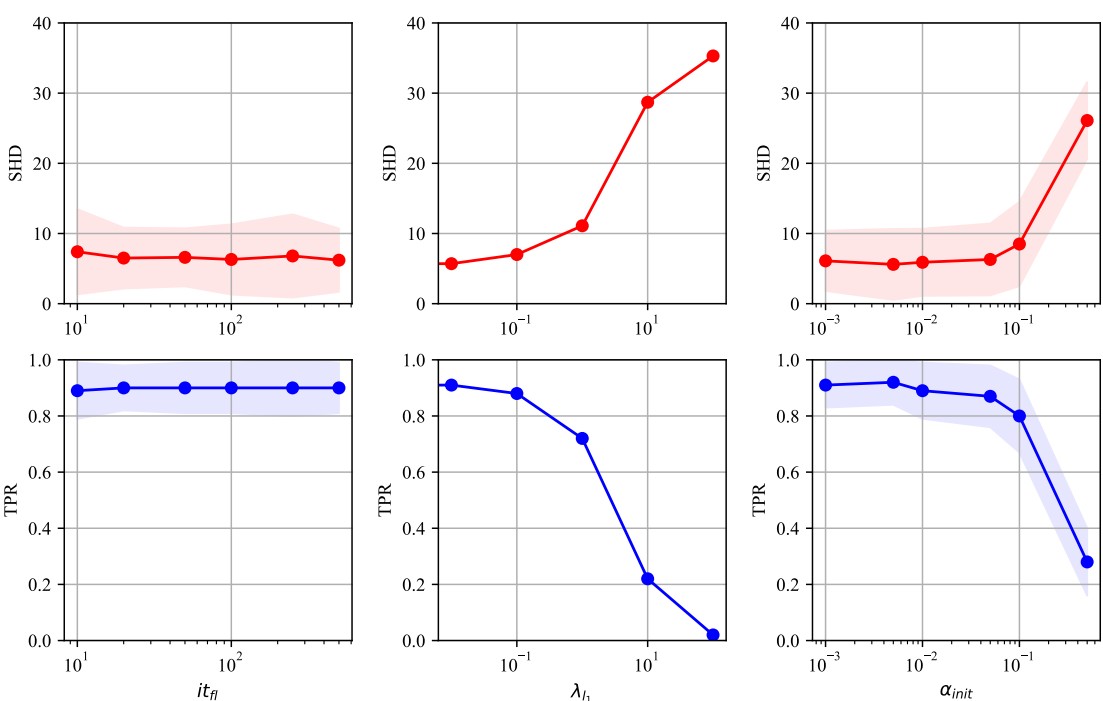

Figure 3: The sensitivity analysis of hyper-parameters

and highly affects the final results. Therefore, carefully picking a proper combination of $\rho_{init}$ and $\beta$ will lead to a better result. In our method, we tune these two parameters via the same scale of experiment with seeds $1 \sim 10$. For each variable scale and training type, the parameters are adjusted once and are applied to all other experiments with the same variable scale. We find the combinations of the following parameters in Table 7 work well in our method. Our method also adopts a $\ell_1$ sparsity term on $g_\tau(\boldsymbol{U})$, where the sparsity coefficient $\lambda_{\ell_1}$ is chosen as 0.01 for all settings.

Table 7: The combinations of $\rho_{init}$ and $\beta$ on simulated data in our method.

| | 10 nodes | | 20 nodes | | 40 nodes | |
|---|---|---|---|---|---|---|
| | $\rho_{init}$ | $\beta$ | $\rho_{init}$ | $\beta$ | $\rho_{init}$ | $\beta$ |
| AS-FCD | $6 \times 10^{-3}$ | 10 | $1 \times 10^{-5}$ | 20 | $1 \times 10^{-11}$ | 120 |
| DS-FCD | $6 \times 10^{-3}$ | 10 | $6 \times 10^{-5}$ | 20 | $1 \times 10^{-11}$ | 120 |

## B.7 Sensitivity analysis of hyper-parameters

Here, we show the sensitivity analysis of $it_{fl}$, $\alpha_{init}$, and $\lambda_{l_1}$. From the experimental results in Figure 3, we find that our method is relatively robust to $it_{fl}$. That is to say, the $it_{fl}$ can be reduced to alleviate the pressure of communication costs while the performance can be well kept. $\lambda_{l_1}$ is the coefficient of $l_1$ sparsity, which will affect the final results. Because we have no sparsity information of the underlying causal graph,

we set $\lambda_{l_1} = 0.01$ in all settings. When dealing with real data, we recommend the audiences adjust this parameter by using our parameter-tuning method provided in the Section B.4. The results of $\alpha_{init}$ are exactly as expected. As discussed before, our method tries to maximize the likelihood term of the total loss in the early stages, which is important to find the final ground-truth DAG. If setting a relatively large $\alpha_{init}$, the early learning stages would be affected. Therefore, we recommend directly taking $\alpha_{init}$ as 0 in all settings.

## C  Discussions on our method

### C.1  Novelty and contributions

Firstly, we acknowledge the contribution of our baseline method MCSL (Ng et al., 2022b), which really performs well in many settings and helps to guarantee the performances of our proposed method. We also appreciate the excellent baseline method FedAvg (McMahan et al., 2017), which provides an efficient federated learning way. Our FCD is highly inspired and benefits from these two works. The main contributions, which can be taken by our proposed method, are (1) **one of the first works** that investigate the practical problem of causal discovery in a federated setup and (2) further providing the DS-FCD approach that can guarantee the **privacy protection** by avoiding the raw data leakage and allow the **data heterogeneity** across the clients. Another concurrent work NOTEARS-ADMM (Ng & Zhang, 2022) also considers the same problem while our DS-FCD can (1) gain better performances in most of the settings, (2) well handle the nonlinear cases, (3) allow non-IID data, and (4) provide a quite flexible federated causal discovery framework.

**Discussions on the simple averaging**  Even though averaging is the simplest way to aggregating and exchanging information, we find it is **quite an effective** way to solve the federated causal discovery problem, which is definitely an advantage of our method. For the IID cases, our simple averaging can nearly approach the same performance as using all data. For the non-IID cases, DS-FCD can still obtain satisfactory results. While, as the future work, more advanced information aggregation methods (Wang et al., 2020a) can be well incorporated into our framework to further boost the performances.

### C.2  Difference with graph neural network (GNN) learning

There are four main reasons, which make CD and GNN two different research lines. (1) Nodes in causal graph represent variables and directed edges describe the cause-effect relation between different variables. In GNN, graph talks more about the graph-type data, such as social networks, protein networks, and traffic networks. (2) Networks in CD are leveraged to learn the causal mechanisms while networks in GNN are taken to achieve node embedding and feature extraction. (3) Learned causal graph can be taken for interventional and counterfactual reasoning. (4) CD cares more about identifiability. That is to say, it is important to exactly identify the true causal process underlying the observations.

### C.3  Broader impact statement

In FL, the server and some clients participate in this process. While as we talked about above, the DAG is shared among all clients. The FCD is motivated by "data on each client is not enough for identifying up the ground-truth DAG". That is to say, the causal graph information is not private for clients. For the server, it depends. In our previous motivations, we actually only care about the "raw data leakage" problem but did not take the privacy of the causal graph into consideration. In real-world scenarios, some of the causal relations can be public such as diseases research. For these cases, our method can still work. However, causality structure sometimes may also be private information. This problem can also be easily solved by picking one client as the proxy server.

### C.4  DS-FCD as a framework

In this paper, we restrict our attention to the case that all concerned variables can be well observed. We also only take MCSL (Ng et al., 2022b) as the baseline method. However, in practice, all gradient-based methods can be incorporated into our AS-FCD framework to deal with the homogeneous/IID data. To deal with the heterogeneous data, we prefer that the baseline methods can separately learn the causal graph

and causal mechanisms. The other baseline methods that can be easily combined into our framework are NOTEARS-MLP (Zheng et al., 2020) and DAG-GNN (Yu et al., 2019). Unfortunately, many works are not in this fashion, such as GraN-DAG (Lachapelle et al., 2020), CD-RL (Zhu et al., 2020) and their following works.

**Latent variables.** In this paper, we carry with causal sufficiency assumption, which assumes that there is no unobserved common confounder. Handling latent confounder is a fundamentally important but really hard problem in the traditional causal discovery not to mention the federated setup. Until now, the theoretical results on the identifiability of causal discovery with latent confounder is always too weak to be used in practice since too strict assumptions are taken. In the recent progress of the latent variables research, Bhattacharya et al. (2021) takes the acyclic directed mixed graphs (ADMGs) to describe the causal graphs with latent confounder. With different types of restrictions, three classes of proprieties named Ancestral graph, Acid graph, and Bow-free graph, are given. According to different proprieties, different graph constraints are given. For example, trace $\left(e^D\right) - d + \text{sum}(D \circ B) = 0$ is set for the Bow-free graph[8], where $D$ is the adjacency matrix recording the directed edges and $B$ records the double-directed edges. To incorporate this method in our framework, we can directly replace the constraint. However, this method can only deal with the linear Gaussian case, which is rather limited.

## C.5 The consistency results by BIC score

Actually, for linear additive noise models with Gaussian noises, the consistency results for maximizing the BIC score to identify the causal graph (Markov Equivalence Class or DAG) have been well established (Tian & Pearl, 2001; Huang et al., 2020a). For this case, with the DAG space constraint, the unique maximum of score function $\mathcal{S}^{c_k}(\mathcal{D}^{c_k}, \mathbf{\Phi}^{c_k}, \boldsymbol{U}^{c_k})$ with BIC score corresponds to the ground-truth DAG. Even for the high-dimensional consistency for linear Gaussian SEM in the case when the model is identifiable (Aragam et al., 2019). Since the ground-truth $\mathcal{G}$ corresponds to each $\mathcal{S}^{c_k}$, the global maximum $\arg\max_{\mathbf{\Phi}, \boldsymbol{U}} \sum_{k=1}^{m} \mathcal{S}^{c_k}(\mathcal{D}^{c_k}, \mathbf{\Phi}^{c_k}, \boldsymbol{U})$ with DAG constraint can lead to the ground-truth causal graph. For nonlinear ANMs, however, even many practical methods, e.g., MCSL (Ng et al., 2022b), NOTEARS-MLP (Zheng et al., 2020), and CD-RL (Zhu et al., 2020), have been proposed to solve this problem by maximizing the BIC score, the theoretical results of consistency is still lacking and would be an interesting future work to be investigated. Therefore, our framework based on these methods inherits the theoretical limit for the nonlinear case. From our paper, however, empirical results can still show the effectiveness of the method.

## C.6 Does the global maximum of Eq. (6) correspond to the ground-truth DAG?

Firstly, for observations of identifiable ANMs on each client, the unique maximum of score function $\mathcal{S}^{c_k}(\mathcal{D}^{c_k}, \mathbf{\Phi}^{c_k}, \boldsymbol{U}^{c_k})$ with BIC score corresponds to the ground-truth DAG (Zheng et al., 2018; Ng et al., 2022b). Even for the high-dimensional consistency for linear Gaussian SEM in the case when the model is identifiable. Since the ground-truth $\mathcal{G}$ corresponds to each $\mathcal{S}^{c_k}$, the global maximum $\arg\max_{\mathbf{\Phi}, \boldsymbol{U}} \sum_{k=1}^{m} \mathcal{S}^{c_k}(\mathcal{D}^{c_k}, \mathbf{\Phi}^{c_k}, \boldsymbol{U}^{c_k})$ with DAG constraint can lead to the ground-truth DAG.

## C.7 Can Algorithms 1 and 2 solve Eq. (6)?

Unfortunately, the global maximum of Eq. (6) can not be well reached by the gradient-based optimization methods, which is mainly caused by the non-convex property of the acyclicity constraint. Firstly, discovering the ground-truth DAG is an NP-hard problem. Traditional methods like PC and GES search the discrete DAG space to solve this problem, which are relatively time-consuming. Then, NOTEARS introduces an equality constraint (3) to formulate the DAG search problem as a continuous optimization problem, which can be easily solved by the gradient descent methods. However, the trade-off is that this equality constraint is non-convex, which pushes us away from finding the ground-truth DAG (the global minima of (6)). That is to say, using gradient descent to solve (6) only can reach the local minima of (6). This similar conclusion stands for recent continuous optimization-based CD methods such as GraNDAG (Lachapelle et al., 2020), DAG-GNN (Yu et al., 2019), and NOTEARS-MLP (Zheng et al., 2020).

---

[8]See more details at Section 4 in (Bhattacharya et al., 2021)

### C.8 Can $U$ finally satisfy the acyclicity constraint in Eq. (3)?

In the following, for simplicity, please do not mind if we explain our method by setting some parameters with specific values. Firstly, following NOTEARS (Zheng et al., 2018) and MCSL (Ng et al., 2022b), we take Augmented Lagrangian Method (ALM) to covert the constrained optimization problem into a series of sub-problems without the hard constraint but with two penalty terms. For the $t$-th sub-problem, the specific formulation of Eq. (8) is related to $\alpha_t$ and $\rho_t$. $\alpha_t$ and $\rho_t$ will be updated to $\alpha_{t+1}$ and $\rho_{t+1}$ after solving the $t$-th sub-problem for 1000 steps (gradient descent step). When dealing with each sub-problem, each client locally updates its personalized model *with acyclicity penalty terms*, which is indeed for the acyclicity constraint. During the 1000 steps, $U$s are averaged every 200 steps (Yes, the simple average is nothing with acyclicity). When finishing 1000 steps (also *the 5-th* 200 *steps* is just finished), a new $U^{new}$ is obtained. Then, $\alpha_t$ and $\rho_t$ are updated to $\alpha_{t+1}$ and $\rho_{t+1}$ to formulate the next sub-problem of ALM, which are described in steps $5 \sim 9$ in Algorithm 1. Then, a new circulation begins. Therefore, we argue that (1) *the acyclicity constraint is guaranteed by taking the acyclicity penalty when solving each sub-problem.* (2) *the convergence of $U$ is supported by the convergence analysis of Personalized FedAvg of Non-IID data.*

### C.9 Convergence Analysis.

This is a new question brought up by our decentralized data setting. Let us quickly review our method. For each client $c_k$, the model parameters include $\boldsymbol{\Phi}^{c_k}$ and $\boldsymbol{U}^{c_k}$. Each client optimizes its parameters on their own data $\mathcal{D}^{c_k}$. The same as NOTEARS and its following works, our method can reach a stationary point instead of the global maximum (the ground-truth DAG). Then, we separate our discussion into two types: IID data and heterogeneous data.

#### C.9.1 IID data.

For IID setup, we have $\boldsymbol{\Phi}^{c_1} = \boldsymbol{\Phi}^{c_2} = \cdots = \boldsymbol{\Phi}^{c_m}$ and $\boldsymbol{U}^{c_1} = \boldsymbol{U}^{c_2} = \cdots = \boldsymbol{U}^{c_m}$. Our method named AS-FCD (All-Shared FCD) sets a central server, which regularly (1) receives all parameters (or $\boldsymbol{U}^{c_k}$ for DS-FCD), (2) averages these parameters to get $\Phi^{new}$ and $U^{new}$ and (3) broadcasts $\Phi^{new}$ and $\boldsymbol{U}^{new}$ to all clients during the learning procedures. AS-FCD benefits from an advanced technique named FedAvg (McMahan et al., 2017) for solving FL problem with IID setting. FedAvg solves the similar problem by averaging all parameters learned from each client in the learning process.

#### C.9.2 Heterogeneous data.

Firstly, to solve the overall constraint-based problem, we take ALM to convert the hard constraint to a soft constraint with a series of increasing penalty co-efficiencies. The convergence of ALM for non-convex problem have been well studied (Nemirovski, 1999) and also be presented in NOTEARS (Zheng et al., 2018). Thus, we only consider the convergence analysis of our method directly from the inner optimization, i.e., the $t$-th sub-problem, as follows.

$$\arg\max_{\boldsymbol{\Phi}, \boldsymbol{U}} \sum_{k=1}^{m} \mathcal{S}^{c_k}\left(\mathcal{D}^{c_k}, \boldsymbol{\Phi}^{c_k}, g_\tau(\boldsymbol{U})\right) - \alpha_t h(\boldsymbol{U}) - \frac{\rho_t}{2} h(\boldsymbol{U})^2, \tag{9}$$

Here, for simplification, we just define that $\hat{\mathcal{S}}^{c_k}(\boldsymbol{\Phi}^{c_k}, \boldsymbol{U}) = -\mathcal{S}^{c_k}\left(\mathcal{D}^{c_k}, \boldsymbol{\Phi}^{c_k}, g_\tau(\boldsymbol{U})\right) + \alpha_t h(\boldsymbol{U}) + \frac{\rho_t}{2} h(\boldsymbol{U})^2$. Then, the overall optimization problem can be reformulated as follows.

$$\arg\min_{\boldsymbol{\Phi}, \boldsymbol{U}} \hat{\mathcal{S}}(\boldsymbol{U}, \boldsymbol{\Phi}) := \sum_{k=1}^{m} \hat{\mathcal{S}}^{c_k}(\boldsymbol{U}, \boldsymbol{\Phi}^{c_k}). \tag{10}$$

Through the following part, we use $\nabla_{\boldsymbol{U}}$ and $\nabla_{\boldsymbol{\Phi}}$ to represent the gradients of $\hat{\mathcal{S}}(\boldsymbol{U}, \boldsymbol{\Phi})$ w.r.t $\boldsymbol{U}$ and $\boldsymbol{\Phi}^{c_k}$, respectively. And, we use $\tilde{\nabla}_{\boldsymbol{U}}$ and $\tilde{\nabla}_{\boldsymbol{\Phi}}$ to represent the stochastic gradients calculated by a mini-batch of observations w.r.t $\boldsymbol{U}$ and $\boldsymbol{\Phi}^{c_k}$, respectively.

**Definition C.1.** (Partial Gradient Diversity). The gradient diversity among all local learning objectives as:

$$\sum_{i=1}^{m} \left\| \nabla_{\boldsymbol{U}} \hat{\mathcal{S}}^{c_k}(\boldsymbol{U}, \boldsymbol{\Phi}^{c_k}) - \nabla_{\boldsymbol{U}} \hat{\mathcal{S}}(\boldsymbol{U}, \boldsymbol{\Phi}) \right\|^2 \leq \delta^2. \tag{11}$$

Note that the notation of gradient diversity is introduced (Yin et al., 2018; Haddadpour & Mahdavi, 2019) as a measurement to compute the similarity among gradients update on different clients.

**Assumption C.2.** (Smoothness and Lower Bound). The local objective function $\hat{\mathcal{S}}^{c_k}(\cdot)$ of the $k$-th client is differentiable for all $k \in [m]$. Also, $\nabla_{\boldsymbol{U}} \hat{\mathcal{S}}^{c_k}(\boldsymbol{U}, \boldsymbol{\Phi}^{c_k})$ is $L_{\boldsymbol{U}}$-Lipschitz w.r.t $\boldsymbol{U}$ and $L_{\boldsymbol{U}\boldsymbol{\Phi}}$ w.r.t $\boldsymbol{\Phi}^{c_k}$, and $\nabla_{\boldsymbol{\Phi}} \hat{\mathcal{S}}^{c_k}(\boldsymbol{U}, \boldsymbol{\Phi}^{c_k})$ is $L_{\boldsymbol{\Phi}}$-Lipschitz w.r.t $\boldsymbol{\Phi}^{c_k}$ and $L_{\boldsymbol{\Phi}\boldsymbol{U}}$ w.r.t $\boldsymbol{U}$. We also assume the overall objective function can be bounded by a constant $\hat{\mathcal{S}}^*$ and denote $\Delta \hat{\mathcal{S}}_0 = \hat{\mathcal{S}}(\boldsymbol{U}_0, \boldsymbol{\Phi}_0) - \hat{\mathcal{S}}^*$.

The relative cross-sensitivity of $\nabla_{\boldsymbol{U}} \hat{\mathcal{S}}^{c_k}$ w.r.t $\boldsymbol{\Phi}^{c_k}$ and $\nabla_{\boldsymbol{\Phi}} \hat{\mathcal{S}}^{c_k}$ w.r.t $\boldsymbol{U}$ with the scalar

$$\chi := \max \left\{ L_{\boldsymbol{U}\boldsymbol{\Phi}}, L_{\boldsymbol{\Phi}\boldsymbol{U}} \right\} / \sqrt{L_{\boldsymbol{U}} L_{\boldsymbol{\Phi}}}. \tag{12}$$

**Assumption C.3.** (Bounded Local Variance) For each local data $\mathcal{D}^{c_k}, k \in [m]$, we can independently sample a batch of data denoted as $\xi \subset \mathcal{D}^{c_k}$. Then, there exist constant $\delta_{\boldsymbol{U}}$ and $\delta_{\boldsymbol{\Phi}}$ such that

$$\mathbf{E} \left[ \left\| \tilde{\nabla}_{\boldsymbol{U}} \hat{\mathcal{S}}^{c_k}(\boldsymbol{\Phi}^{c_k}, \boldsymbol{U}) - \nabla_{\boldsymbol{U}} \hat{\mathcal{S}}^{c_k}(\boldsymbol{U}, \boldsymbol{\Phi}^{c_k}) \right\|^2 \right] \leq \sigma_{\boldsymbol{U}}^2,$$

$$\mathbf{E} \left[ \left\| \tilde{\nabla}_{\boldsymbol{\Phi}} \hat{\mathcal{S}}^{c_k}(\boldsymbol{\Phi}^{c_k}, \boldsymbol{U}) - \nabla_{\boldsymbol{\Phi}} \hat{\mathcal{S}}^{c_k}(\boldsymbol{U}, \boldsymbol{\Phi}^{c_k}) \right\|^2 \right] \leq \sigma_{\boldsymbol{\Phi}}^2,$$

The bounded variance assumption is a standard assumption on the stochastic gradients (Haddadpour & Mahdavi, 2019; Pillutla et al., 2022).

**Theorem C.4.** *(Convergence of DS-FCD). For DS-FCD with all clients involved in the aggregation, for all $0 \leq it \leq T - 1$, under Assumptions C.2, C.3 and C.1, and the learning rate for the $\boldsymbol{U}$ part is set as $\eta/(L_{\boldsymbol{U}} it_{in})$ and the learning rate for the $\phi$ part is set as $\eta/(L_{\phi} it_{in})$. Then, for $\eta$ depending on the problem parameters, we have*

$$\frac{1}{T} \sum_{it=0}^{T-1} \left( \frac{1}{L_{\boldsymbol{U}}} \mathbb{E} \left[ \left\| \nabla_{\boldsymbol{U}} \hat{\mathcal{S}}^{c_k}(\boldsymbol{\Phi}_{it}^{c_k}, \boldsymbol{U}_{it}) \right\|^2 \right] + \frac{1}{L_{\boldsymbol{\Phi}}} \mathbb{E} \left[ \frac{1}{m} \sum_{i=1}^{m} \left\| \nabla_{\boldsymbol{U}} \hat{\mathcal{S}}^{c_k}(\boldsymbol{\Phi}_{it}^{c_k}, \boldsymbol{U}_{it}) \right\|^2 \right] \right) \leq$$
$$\frac{(\Delta \hat{\mathcal{S}}_0 \sigma_{\text{fcd},1}^2)^{1/2}}{\sqrt{T}} + \frac{(\Delta \hat{\mathcal{S}}_0^2 \sigma_{\text{fcd},2}^2)^{1/3}}{T^{2/3}} + \mathcal{O}(\frac{1}{T}). \tag{13}$$

*where we define the effective variance terms*

$$\sigma_{\text{fcd},1}^2 = \left( 1 + \chi^2 \right) \left( \frac{\sigma_{\boldsymbol{U}}^2}{L_{\boldsymbol{U}}} + \frac{\sigma_{\boldsymbol{\Phi}}^2}{L_{\boldsymbol{\Phi}}} \right),$$
$$\sigma_{\text{fcd},2}^2 = \left( 1 + \chi^2 \right) \left( \frac{\delta^2}{L_{\boldsymbol{U}}} + \frac{\sigma_{\boldsymbol{U}}^2}{L_{\boldsymbol{U}}} + \frac{\sigma_{\boldsymbol{\Phi}}^2}{L_{\boldsymbol{\Phi}}} \right) \left( 1 - \frac{1}{it_{in}} \right), \tag{14}$$

*where $it_{in}$ is the total step of one inner loop used in lines $4 - 21$ in Algorithm 2.*

From Theorem C.4, we can see that the gradients $\nabla_{\boldsymbol{U}} \hat{\mathcal{S}}^{c_k}(\boldsymbol{\Phi}_{it}^{c_k}, \boldsymbol{U}_{it})$ w.r.t $\boldsymbol{U}$ and $\nabla_{\boldsymbol{U}} \hat{\mathcal{S}}^{c_k}(\boldsymbol{\Phi}_{it}^{c_k}, \boldsymbol{U}_{it})$ w.r.t $\boldsymbol{\Phi}$ at the $t$-th step can be bounded if we choose a proper $\eta$, which affects the learning rates of the model.

The proof of Theorem C.4 can be borrowed from the proof of Theorem 2 in (Pillutla et al., 2022). Notice that, in our theorem, we have assumed that all clients participate the aggregation for simplification and the conclusion can be easily extended to the general partial participation case.

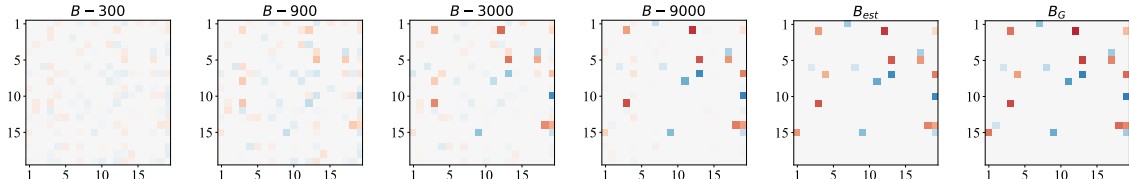

Figure 4: Visualization of the learned graph during the optimization process. $B - n$ means the learned graph in the $n$ steps. $B_{est}$ is the final estimated causal graph. $B_G$ is the ground-truth DAG.

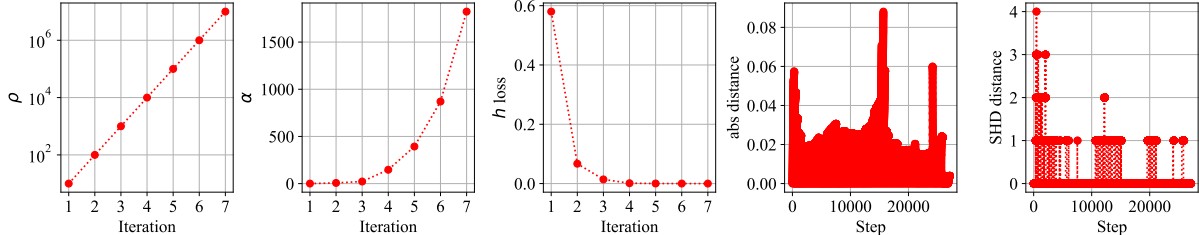

Figure 5: Parameters changing during the optimization process. The first three sub-figures include the changes of penalty coefficients $rho$, $alpha$ and the DAG constraint loss $h_{loss}$. The fourth sub-figure records the $\ell_1$ distance between two learned graphs on the different clients. The fifth sub-figure records the SHD distance between two learned graphs on the different clients.

## D   Supplementary experimental details

### D.1   Visualization of the learned DAG of FCD

**Visualization of the learned DAG of FCD.** We take an example of the AS-FCD optimization process on linear Gaussian model with NOTEARS as the baseline method and plot the change in estimated parameters in Fig. 4 and Fig. 5. In this example, the number of nodes is set as 10 and the edges are 10. The data is simulated by ER graph and evenly assign 200 observations on two different clients. In Fig. 4, we can see that the learned causal graph is asymptotically approximates the ground-truth DAG $B_G$, including the existence of edges and their weights. From Fig. 5, we can find that, with the increase of the penalty coefficients, the $h_{loss}$ deceases quickly. For learned graphs on the different clients, we can see that the SHD distance is smaller during the optimization procedures.

### D.2   Uneven distributions

For federated learning problems in real world, different clients may own different amounts of observations. To verify the stability of our method, we simulate the setting that uneven distributions in different clients. For each client, the number of observations are randomly chosen from a list $[20\%, 40\%, 60\%, 80\%] \times n$, where $n$ is the maximal observations. The experimental results are shown in Fig. 6, from which we can find that our method show relatively stable performance in this setting.

### D.3   Varying clients

In this setting, we now consider a fixed number of samples which are distributed across different number of clients. We conduct experiments for $(2, 4, 6, 8)$ clients and show the results in Fig. 7. With the increase of clients number, our method can show better performances.

### D.4   Dense graphs

Our method is also implemented on some denser graphs. Experimental results in Table 8 and Table 9. From these experimental results, we can see that our method shows consistently better performance over other

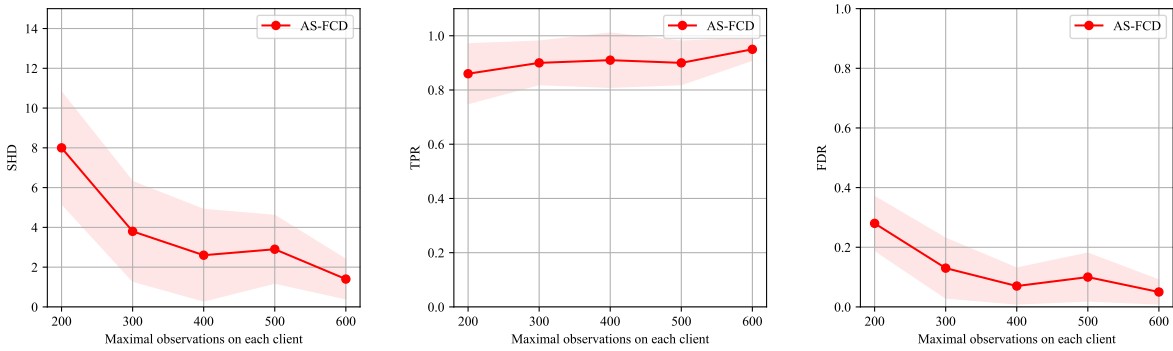

Figure 6: Results of uneven distributions on different clients.

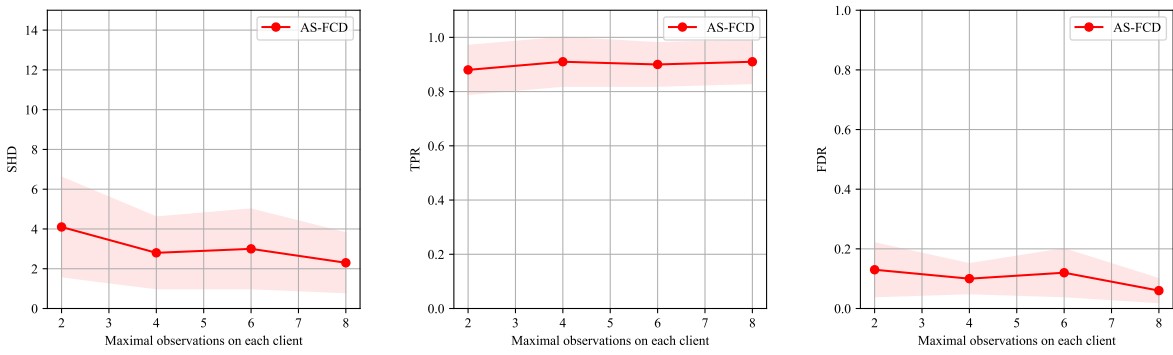

Figure 7: Results of performances with varying clients.

methods on the denser graph setting. For the IID case, both AS-DAG and DS-DAG obtain the nearly low SHD as MCSL trained on all data and far better than all methods trained on separated data. For the Non-IID case, our DS-FCD still shows the best performance. Compared to NOTEARS in 20 variables case, DS-FCD shows similar SHD results but much better TPR result. Therefore, how to reduce the false discovery rate of DS-FCD would be an interesting thing.

Table 8: Results on nonlinear ANM with dense graphs (IID).

| | | ER4 with 10 nodes | | SF4 with 10 nodes | | ER4 with 20 nodes | | SF4 with 20 nodes | |
|---|---|---|---|---|---|---|---|---|---|
| | | SHD ↓ | TPR ↑ | SHD ↓ | TPR ↑ | SHD ↓ | TPR ↑ | SHD ↓ | TPR ↑ |
| All data | PC | $27.3 \pm 3.2$ | $0.29 \pm 0.07$ | $18.9 \pm 4.9$ | $0.37 \pm 0.16$ | $68.2 \pm 9.5$ | $0.23 \pm 0.06$ | $60.2 \pm 9.3$ | $0.30 \pm 0.08$ |
| | NOTEARS | $34.3 \pm 1.7$ | $0.03 \pm 0.02$ | $22.7 \pm 1.3$ | $0.05 \pm 0.05$ | $71.8 \pm 7.2$ | $0.03 \pm 0.01$ | $62.8 \pm 0.9$ | $0.02 \pm 0.01$ |
| | MCSL | $15.5 \pm 5.9$ | $0.57 \pm 0.15$ | $4.5 \pm 3.1$ | $0.83 \pm 0.11$ | $33.8 \pm 10.4$ | $0.55 \pm 0.11$ | $19.8 \pm 7.5$ | $0.69 \pm 0.11$ |
| Sep data | PC | $31.5 \pm 2.1$ | $0.14 \pm 0.03$ | $20.4 \pm 0.58$ | $0.21 \pm 0.03$ | $68.7 \pm 8.1$ | $0.13 \pm 0.03$ | $60.9 \pm 2.8$ | $0.15 \pm 0.02$ |
| | NOTEARS | $34.3 \pm 1.8$ | $0.03 \pm 0.01$ | $22.7 \pm 1.0$ | $0.06 \pm 0.04$ | $70.1 \pm 6.9$ | $0.03 \pm 0.01$ | $62.3 \pm 0.56$ | $0.03 \pm 0.01$ |
| | MCSL | $\mathbf{15.8 \pm 3.3}$ | $\mathbf{0.61 \pm 0.09}$ | $8.3 \pm 4.3$ | $\mathbf{0.78 \pm 0.11}$ | $49.3 \pm 11.8$ | $\mathbf{0.63 \pm 0.10}$ | $39.7 \pm 5.6$ | $\mathbf{0.73 \pm 0.07}$ |
| | DS-FCD | $\mathbf{16.9 \pm 4.9}$ | $\mathbf{0.53 \pm 0.12}$ | $\mathbf{5.4 \pm 3.0}$ | $0.78 \pm 0.12$ | $\mathbf{35.4 \pm 10.9}$ | $0.53 \pm 0.11$ | $\mathbf{20.7 \pm 5.1}$ | $0.69 \pm 0.08$ |
| | AS-FCD | $17.4 \pm 4.8$ | $\mathbf{0.53 \pm 0.12}$ | $\mathbf{5.5 \pm 2.8}$ | $\mathbf{0.79 \pm 0.11}$ | $40.7 \pm 4.8$ | $\mathbf{0.57 \pm 0.10}$ | $24.1 \pm 5.8$ | $\mathbf{0.71 \pm 0.09}$ |

## D.5 Voting method.

There is another interesting research line (Na & Yang, 2010), which also try to learn DAG from decentralized data. We add a DAG combination method proposed in (Na & Yang, 2010), which proposes to vote for each entry of the adjacency matrix to get the final DAG. From the experimental results in Table 10, we can find that For PC and NOTEARS, the combining method seems to contribute little improvement. This is because

Table 9: Results on nonlinear ANM with dense graphs (Non-IID).

| | | ER4 with 10 nodes | | SF4 with 10 nodes | | ER4 with 20 nodes | | SF4 with 20 nodes | |
| | | SHD ↓ | TPR ↑ | SHD ↓ | TPR ↑ | SHD ↓ | TPR ↑ | SHD ↓ | TPR ↑ |
|---|---|---|---|---|---|---|---|---|---|
| Sep data | PC | $29.3 \pm 1.3$ | $0.23 \pm 0.03$ | $20.3 \pm 2.1$ | $0.31 \pm 0.06$ | $71.9 \pm 8.1$ | $0.19 \pm 0.03$ | $62.7 \pm 2.8$ | $0.22 \pm 0.03$ |
| | NOTEARS | $20.5 \pm 2.6$ | $0.45 \pm 0.08$ | $12.2 \pm 2.9$ | $0.54 \pm 0.11$ | $43.2 \pm 7.0$ | $0.49 \pm 0.08$ | $\mathbf{39.4 \pm 6.8}$ | $0.47 \pm 0.10$ |
| | MCSL | $20.0 \pm 3.2$ | $0.52 \pm 0.07$ | $13.7 \pm 2.2$ | $0.65 \pm 0.07$ | $65.1 \pm 7.7$ | $0.33 \pm 0.05$ | $59.4 \pm 5.3$ | $0.31 \pm 0.05$ |
| | DS-FCD | $\mathbf{8.5 \pm 3.7}$ | $\mathbf{0.84 \pm 0.09}$ | $4.5 \pm 2.0$ | $\mathbf{0.93 \pm 0.07}$ | $\mathbf{40.7 \pm 14.5}$ | $\mathbf{0.74 \pm 0.07}$ | $39.9 \pm 10.8$ | $\mathbf{0.68 \pm 0.07}$ |

the reported DAGs local clients are too bad to get a good result. For MCSL, this combing method works really well for improving the performance. The reason is easy to be inferred from the results. For MCSL, DAGs reported by local clients are of bad SHDs but good TPR, which means that the False Discovery Rates (FDRs) are high. While the combing method can further reduce the FDRs and keep the TPRs still good. Then, SHD can be further reduced. Luckily, our DS-FCD still shows the best performances in all settings.

Table 10: Comparison with the voting method.

| | | IID-GP | | | | Non-IID | | | |
| | | ER2 with 10 nodes | | ER2 with 20 nodes | | ER2 with 10 nodes | | ER2 with 20 nodes | |
| | | SHD ↓ | TPR ↑ | SHD ↓ | TPR ↑ | SHD ↓ | TPR ↑ | SHD ↓ | TPR ↑ |
|---|---|---|---|---|---|---|---|---|---|
| Sep data | PC | $14.1 \pm 2.4$ | $0.31 \pm 0.06$ | $32.7 \pm 6.5$ | $0.28 \pm 0.07$ | $12.5 \pm 2.7$ | $0.45 \pm 0.07$ | $28.5 \pm 6.3$ | $0.44 \pm 0.07$ |
| | NOTEARS | $16.5 \pm 2.0$ | $0.06 \pm 0.04$ | $31.7 \pm 6.0$ | $0.11 \pm 0.04$ | $7.6 \pm 2.6$ | $0.60 \pm 0.11$ | $15.0 \pm 3.1$ | $0.62 \pm 0.09$ |
| | MCSL | $7.1 \pm 3.2$ | $0.83 \pm 0.08$ | $24.8 \pm 5.5$ | $0.88 \pm 0.07$ | $9.2 \pm 1.8$ | $0.72 \pm 0.06$ | $23.3 \pm 5.8$ | $0.56 \pm 0.08$ |
| Voting | PC | $13.3 \pm 3.0$ | $0.27 \pm 0.11$ | $29.7 \pm 5.9$ | $0.22 \pm 0.05$ | $11.4 \pm 3.4$ | $0.36 \pm 0.13$ | $25.5 \pm 6.8$ | $0.29 \pm 0.13$ |
| | NOTEARS | $15.6 \pm 2.2$ | $0.11 \pm 0.06$ | $32.6 \pm 6.2$ | $0.09 \pm 0.05$ | $7.8 \pm 4.0$ | $0.56 \pm 0.20$ | $18.4 \pm 11.6$ | $0.49 \pm 0.30$ |
| | MCSL | $8.0 \pm 3.1$ | $0.85 \pm 0.16$ | $18.1 \pm 7.8$ | $\mathbf{0.88 \pm 0.06}$ | $6.9 \pm 2.2$ | $0.71 \pm 0.13$ | $10.1 \pm 4.6$ | $0.79 \pm 0.09$ |
| | DS-FCD | $\mathbf{2.4 \pm 2.0}$ | $\mathbf{0.86 \pm 0.12}$ | $6.2 \pm 4.0$ | $0.85 \pm 0.10$ | $\mathbf{1.9 \pm 1.6}$ | $\mathbf{0.99 \pm 0.02}$ | $6.2 \pm 4.7$ | $\mathbf{0.89 \pm 0.09}$ |
| | AS-FCD | $\mathbf{1.8 \pm 2.0}$ | $\mathbf{0.89 \pm 0.12}$ | $\mathbf{5.0 \pm 4.2}$ | $\mathbf{0.88 \pm 0.11}$ | NaN | NaN | NaN | NaN |

## D.6 Comparisons with CAM

Here, we add one more identifiable baseline named causal additive model (CAM) (Bühlmann et al., 2014), which also serves as a baseline in MCSL (Ng et al., 2022b), GraNDAG (Lachapelle et al., 2020), and DAG-GAN (Yu et al., 2019). From result in Table 11 and 12, we can see that our methods always show an advantage over CAM. CAM also assumes a non-linear additive noise model for data generation. However, CAM limits the non-linear function to be additive. In normal ANM, $X_i = f_i(X_{pa_i}) + \epsilon_i$ while CAM assumes $X_i = \sum_{j \in X(pa_i)} f_{i \leftarrow j}(X_j) + \epsilon_i$, which limits the capacity of its model. From the above experimental results, we can see that our methods show consistent advantages over CAM.

Table 11: Comparisons with CAM on nonlinear ANM (IID-GP).

| | | ER2 with 10 nodes | | SF2 with 10 nodes | | ER2 with 20 nodes | | SF2 with 20 nodes | |
| | | SHD ↓ | TPR ↑ | SHD ↓ | TPR ↑ | SHD ↓ | TPR ↑ | SHD ↓ | TPR ↑ |
|---|---|---|---|---|---|---|---|---|---|
| All data | CAM | $9.5 \pm 2.9$ | $0.87 \pm 0.09$ | $9.1 \pm 3.1$ | $0.84 \pm 0.10$ | $21.4 \pm 4.7$ | $0.77, \pm 0.08$ | $26.6 \pm 6.1$ | $0.75 \pm 0.07$ |
| Sep data | CAM | $11.8 \pm 2.6$ | $0.40 \pm 0.10$ | $11.1 \pm 1.5$ | $0.38 \pm 0.11$ | $24.3 \pm 5.8$ | $0.40 \pm 0.07$ | $26.8 \pm 2.0$ | $0.36 \pm 0.06$ |
| | DS-FCD | $2.4 \pm 2.0$ | $0.86 \pm 0.12$ | $2.7 \pm 2.2$ | $\mathbf{0.86 \pm 0.13}$ | $6.2 \pm 4.0$ | $0.85 \pm 0.10$ | $14.7 \pm 7.0$ | $\mathbf{0.80 \pm 0.11}$ |
| | AS-FCD | $\mathbf{1.8 \pm 2.0}$ | $\mathbf{0.89 \pm 0.12}$ | $\mathbf{2.5 \pm 2.7}$ | $0.85 \pm 0.15$ | $\mathbf{5.0 \pm 4.2}$ | $\mathbf{0.88 \pm 0.11}$ | $\mathbf{7.8 \pm 5.5}$ | $0.80 \pm 0.14$ |

## D.7 Comparisons with NOTEARS-ADMM

In this subsection, we give the experimental comparisons with NOTEARS-ADMM in detail to verify the advantage of our averaging strategy is simple but effective. Firstly, we conduct the results on linear models,

Table 12: Comparisons with CAM on nonlinear ANM (Non-IID).

| | | ER2 with 10 nodes | | SF2 with 10 nodes | | ER2 with 20 nodes | | SF2 with 20 nodes | |
|---|---|---|---|---|---|---|---|---|---|
| | | SHD ↓ | TPR ↑ | SHD ↓ | TPR ↑ | SHD ↓ | TPR ↑ | SHD ↓ | TPR ↑ |
| All data | CAM | $31.9 \pm 4.8$ | $0.39 \pm 0.15$ | $31.8 \pm 4.4$ | $0.31 \pm 0.17$ | $104.6 \pm 15.4$ | $0.46 \pm 0.15$ | $116.9 \pm 13.8$ | $0.35 \pm 0.07$ |
| Sep data | CAM | $18.0 \pm 1.7$ | $0.52 \pm 0.04$ | $17.8 \pm 2.1$ | $0.51 \pm 0.3$ | $47.5 \pm 9.2$ | $0.52 \pm 0.04$ | $53.0 \pm 6.1$ | $0.50 \pm 0.03$ |
| | DS-FCD | $\mathbf{1.9 \pm 1.6}$ | $\mathbf{0.99 \pm 0.02}$ | $\mathbf{2.6 \pm 1.3}$ | $\mathbf{0.93 \pm 0.07}$ | $\mathbf{6.2 \pm 4.7}$ | $\mathbf{0.89 \pm 0.09}$ | $\mathbf{11.5 \pm 6.7}$ | $\mathbf{0.81 \pm 0.14}$ |

which are the main part in Ng & Zhang (2022). As shown in Fig. 8, even on linear models, our AS-FCD can consistently show its advantage over NOTEARS-ADMM. Then, for the nonlinear models, we consider two different functions named MLP and Gaussian process (GP). The results are presented in Fig. 9, from which we can see that FCD always show better performance over all settings. Since NOTEARS-ADMM can not handle heterogeneous data, we do not give the results on Non-IID data for fair comparison.

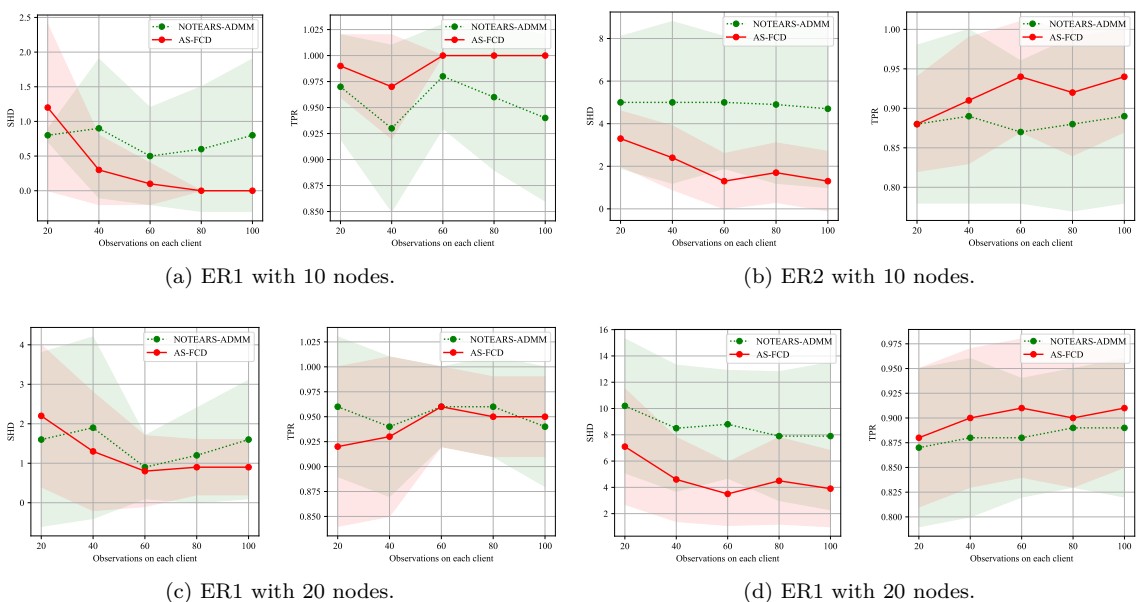

(a) ER1 with 10 nodes.

(b) ER2 with 10 nodes.

(c) ER1 with 20 nodes.

(d) ER1 with 20 nodes.

Figure 8: Comparisons with NOTEARS-ADMM on linear model (IID).

Table 13: Results on nonlinear ANM with different functions (IID, 10 nodes, ER2).

| | | GP | | MIM | | MLP | | GP-add | |
|---|---|---|---|---|---|---|---|---|---|
| | | SHD ↓ | TPR ↑ | SHD ↓ | TPR ↑ | SHD ↓ | TPR ↑ | SHD ↓ | TPR ↑ |
| All data | PC | $15.3 \pm 2.6$ | $0.37 \pm 0.10$ | $11.0 \pm 4.9$ | $0.60 \pm 0.16$ | $11.8 \pm 4.3$ | $0.61 \pm 0.14$ | $14.0 \pm 4.7$ | $0.49 \pm 0.16$ |
| | GES | $13.0 \pm 3.9$ | $0.50 \pm 0.18$ | $9.6 \pm 4.4$ | $0.71 \pm 0.17$ | $15.8 \pm 6.0$ | $0.63 \pm 0.14$ | $14.4 \pm 4.9$ | $0.57 \pm 0.17$ |
| | DAG-GNN | $16.2 \pm 2.1$ | $0.07 \pm 0.06$ | $13.7 \pm 2.4$ | $0.26 \pm 0.10$ | $18.2 \pm 3.3$ | $0.36 \pm 0.12$ | $13.3 \pm 2.3$ | $0.24 \pm 0.10$ |
| | NOTEARS | $16.5 \pm 2.0$ | $0.05 \pm 0.04$ | $12.1 \pm 3.2$ | $0.34 \pm 0.13$ | $13.3 \pm 3.4$ | $0.35 \pm 0.15$ | $13.4 \pm 2.2$ | $0.23 \pm 0.09$ |
| | N-S-MLP | $8.1 \pm 3.8$ | $0.56 \pm 0.17$ | $1.6 \pm 1.3$ | $0.95 \pm 0.06$ | $\mathit{5.6 \pm 1.3}$ | $\mathit{0.81 \pm 0.11}$ | $6.8 \pm 4.0$ | $0.65 \pm 0.16$ |
| | MCSL | $\mathit{1.9 \pm 1.5}$ | $\mathit{0.90 \pm 0.08}$ | $\mathit{0.7 \pm 1.2}$ | $\mathit{0.97 \pm 0.06}$ | $12.7 \pm 3.6$ | $0.58 \pm 0.24$ | $\mathit{1.9 \pm 1.7}$ | $\mathit{0.91 \pm 0.07}$ |
| Sep data | PC | $14.1 \pm 2.4$ | $0.31 \pm 0.06$ | $11.1 \pm 3.6$ | $0.48 \pm 0.14$ | $13.2 \pm 3.6$ | $0.42 \pm 0.09$ | $13.5 \pm 3.2$ | $0.37 \pm 0.12$ |
| | GES | $12.7 \pm 2.7$ | $0.37 \pm 0.09$ | $10.6 \pm 3.3$ | $0.54 \pm 0.12$ | $14.6 \pm 4.6$ | $0.50 \pm 0.13$ | $12.0 \pm 2.6$ | $0.48 \pm 0.08$ |
| | DAG-GNN | $15.7 \pm 2.3$ | $0.11 \pm 0.05$ | $11.7 \pm 3.3$ | $0.37 \pm 0.12$ | $17.7 \pm 3.6$ | $0.39 \pm 0.11$ | $13.0 \pm 2.0$ | $0.26 \pm 0.10$ |
| | NOTEARS | $16.5 \pm 2.0$ | $0.06 \pm 0.04$ | $12.3 \pm 3.0$ | $0.33 \pm 0.12$ | $13.4 \pm 3.4$ | $0.35 \pm 0.14$ | $13.3 \pm 2.3$ | $0.24 \pm 0.09$ |
| | N-S-MLP | $8.5 \pm 2.9$ | $0.56 \pm 0.13$ | $2.8 \pm 1.5$ | $\mathbf{0.93 \pm 0.06}$ | $\mathbf{6.4 \pm 1.3}$ | $\mathbf{0.81 \pm 0.11}$ | $7.4 \pm 2.9$ | $0.67 \pm 0.13$ |
| | MCSL | $7.1 \pm 3.2$ | $0.83 \pm 0.08$ | $4.4 \pm 2.1$ | $\mathbf{0.91 \pm 0.06}$ | $13.4 \pm 3.9$ | $0.57 \pm 0.21$ | $6.5 \pm 3.5$ | $0.84 \pm 0.07$ |
| | DS-FCD | $\mathbf{2.4 \pm 2.0}$ | $\mathbf{0.86 \pm 0.12}$ | $\mathbf{2.1 \pm 1.4}$ | $0.91 \pm 0.07$ | $11.1 \pm 3.1$ | $0.57 \pm 0.20$ | $\mathbf{2.6 \pm 1.6}$ | $\mathbf{0.87 \pm 0.09}$ |
| | AS-FCD | $\mathbf{1.8 \pm 2.0}$ | $\mathbf{0.89 \pm 0.12}$ | $\mathbf{1.7 \pm 1.6}$ | $0.91 \pm 0.08$ | $\mathbf{10.5 \pm 3.5}$ | $\mathbf{0.59 \pm 0.22}$ | $\mathbf{2.4 \pm 1.6}$ | $\mathbf{0.87 \pm 0.08}$ |

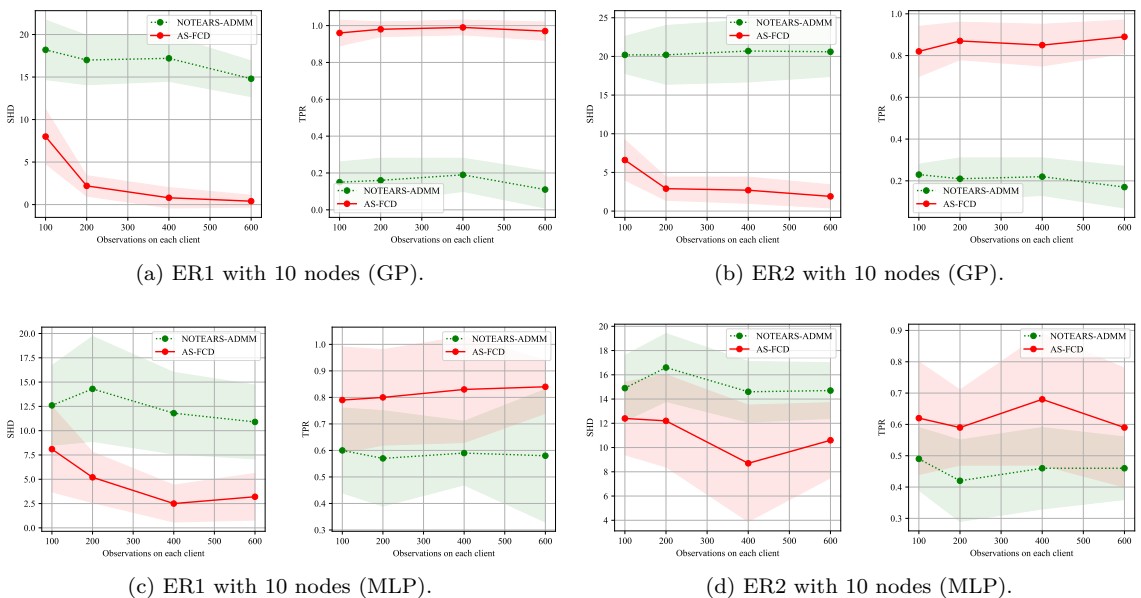

(a) ER1 with 10 nodes (GP).                    (b) ER2 with 10 nodes (GP).

(c) ER1 with 10 nodes (MLP).                   (d) ER2 with 10 nodes (MLP).

Figure 9: Comparisons with NOTEARS-ADMM on nonlinear models (IID).

Table 14: Results on nonlinear ANM with different functions (IID, 20 nodes, ER2).

|  |  | GP | | MIM | | MLP | | GP-add | |
|---|---|---|---|---|---|---|---|---|---|
|  |  | SHD ↓ | TPR ↑ | SHD ↓ | TPR ↑ | SHD ↓ | TPR ↑ | SHD ↓ | TPR ↑ |
| All data | PC | $32.7 \pm 9.4$ | $0.48 \pm 0.13$ | $22.8 \pm 5.8$ | $0.60 \pm 0.15$ | $33.7 \pm 12.3$ | $0.50 \pm 0.13$ | $35.2 \pm 8.0$ | $0.50 \pm 0.09$ |
|  | GES | $27.1 \pm 8.5$ | $0.56 \pm 0.11$ | $21.5 \pm 6.1$ | $0.78 \pm 0.09$ | $44.9 \pm 12.5$ | $0.65 \pm 0.11$ | $41.7 \pm 11.6$ | $0.66 \pm 0.08$ |
|  | DAG-GNN | $32.5 \pm 6.8$ | $0.10 \pm 0.08$ | $26.7 \pm 7.4$ | $0.26 \pm 0.13$ | $32.1 \pm 10.4$ | $0.38 \pm 0.08$ | $27.2 \pm 2.4$ | $0.24 \pm 0.08$ |
|  | NOTEARS | $31.8 \pm 6.0$ | $0.11 \pm 0.04$ | $25.6 \pm 6.1$ | $0.29 \pm 0.08$ | $25.3 \pm 8.0$ | $0.40 \pm 0.09$ | $25.6 \pm 3.9$ | $0.28 \pm 0.06$ |
|  | N-S-MLP | $18.2 \pm 4.5$ | $0.52 \pm 0.10$ | $4.1 \pm 2.0$ | $0.95 \pm 0.04$ | $8.0 \pm 3.9$ | $0.86 \pm 0.07$ | $12.6 \pm 2.2$ | $0.70 \pm 0.06$ |
|  | MCSL | $4.6 \pm 4.6$ | $0.90 \pm 0.13$ | $1.7 \pm 1.6$ | $0.97 \pm 0.04$ | $18.1 \pm 6.6$ | $0.72 \pm 0.14$ | $3.1 \pm 1.9$ | $0.92 \pm 0.05$ |
| Sep data | PC | $32.7 \pm 6.5$ | $0.28 \pm 0.07$ | $24.4 \pm 5.6$ | $0.46 \pm 0.11$ | $30.6 \pm 8.0$ | $0.41 \pm 0.09$ | $29.5 \pm 5.6$ | $0.42 \pm 0.10$ |
|  | GES | $28.6 \pm 5.5$ | $0.34 \pm 0.06$ | $20.5 \pm 3.7$ | $0.61 \pm 0.06$ | $34.4 \pm 11.3$ | $0.52 \pm 0.09$ | $29.3 \pm 5.5$ | $0.51 \pm 0.07$ |
|  | DAG-GNN | $31.7 \pm 6.1$ | $0.12 \pm 0.04$ | $26.8 \pm 5.8$ | $0.26 \pm 0.06$ | $34.1 \pm 9.7$ | $0.46 \pm 0.07$ | $26.5 \pm 4.0$ | $0.27 \pm 0.05$ |
|  | NOTEARS | $31.7 \pm 6.0$ | $0.11 \pm 0.04$ | $25.7 \pm 5.9$ | $0.29 \pm 0.07$ | $25.4 \pm 7.4$ | $0.42 \pm 0.07$ | $25.6 \pm 3.8$ | $0.29 \pm 0.06$ |
|  | N-S-MLP | $19.5 \pm 4.7$ | $0.52 \pm 0.07$ | $6.5 \pm 1.9$ | $0.92 \pm 0.03$ | $16.1 \pm 8.6$ | $0.86 \pm 0.07$ | $16.2 \pm 3.3$ | $0.70 \pm 0.07$ |
|  | MCSL | $24.8 \pm 5.5$ | $0.88 \pm 0.07$ | $20.4 \pm 3.8$ | $0.91 \pm 0.05$ | $30.2 \pm 5.1$ | $0.67 \pm 0.12$ | $16.2 \pm 5.3$ | $0.87 \pm 0.05$ |
|  | DS-FCD | $6.2 \pm 4.0$ | $0.85 \pm 0.10$ | $8.5 \pm 2.8$ | $0.93 \pm 0.05$ | $21.4 \pm 7.9$ | $0.71 \pm 0.14$ | $8.1 \pm 3.2$ | $0.85 \pm 0.05$ |
|  | AS-FCD | $5.0 \pm 4.2$ | $0.88 \pm 0.11$ | $3.3 \pm 2.5$ | $0.92 \pm 0.07$ | $20.1 \pm 8.3$ | $0.72 \pm 0.14$ | $5.6 \pm 2.8$ | $0.86 \pm 0.06$ |

# E  More discussions on the experimental results

## E.1  Why do baseline methods perform not well?

Here, we give the detailed discussions on the experimental results in the paper. First of all, PC and GES can only reach the CPDAG (or MEC) at most, which shares the same skeleton with the ground-truth DAG. When we evaluate SHD, we just ignore the direction of undirected edges learned by PC and GES. That is to say, these two methods can get SHD 0 if they can identify the CPDAG. Therefore, the final results are not caused by unfair comparison. For PC, the independence test is leveraged to decode the (conditional) independence from the data distribution. Therefore, the accuracy would be affected by (1) the amount of the observations and (2) the effectiveness of *the non-parametric kernel independence test* method. GES leverages greedy search with BIC score. However, the likelihood part of BIC in GES is Linear Gaussian, which is unsuitable for data generated by the Non-linear model. NOTEARS is a linear model but the causal mechanisms are non-linear. The reason will be the unfitness between data and model. Therefore, the comparisons with GES

Table 15: Results on Non-IID setting with the different number of observations, (20nodes, ER2).

| | | n =100 | | n =300 | | n =600 | | n =900 | |
|---|---|---|---|---|---|---|---|---|---|
| | | SHD ↓ | TPR ↑ | SHD ↓ | TPR ↑ | SHD ↓ | TPR ↑ | SHD ↓ | TPR ↑ |
| All data | PC | $55.5 \pm 8.5$ | $0.21 \pm 0.06$ | $57.3 \pm 5.7$ | $0.29 \pm 0.07$ | $60.4 \pm 9.8$ | $0.32 \pm 0.11$ | $62.4 \pm 6.6$ | $0.29 \pm 0.10$ |
| | GES | $82.8 \pm 13.7$ | $0.38 \pm 0.12$ | $96.4 \pm 14.9$ | $0.48 \pm 0.08$ | $102.9 \pm 13.6$ | $0.51 \pm 0.08$ | $106.3 \pm 14.3$ | $0.50 \pm 0.11$ |
| | DAG-GNN | $61.8 \pm 14.7$ | $0.39 \pm 0.07$ | $56.8 \pm 9.7$ | $0.37 \pm 0.08$ | $57.7 \pm 12.0$ | $0.38 \pm 0.08$ | $57.9 \pm 12.1$ | $0.32 \pm 0.08$ |
| | NOTEARS | $58.7 \pm 12.8$ | $0.41 \pm 0.12$ | $57.6 \pm 10.2$ | $0.44 \pm 0.06$ | $57.3 \pm 12.9$ | $0.43 \pm 0.08$ | $59.4 \pm 10.3$ | $0.39 \pm 0.10$ |
| | N-S-MLP | $111.2 \pm 14.4$ | $0.92 \pm 0.10$ | $101.0 \pm 16.8$ | $0.92 \pm 0.05$ | $100.8 \pm 14.7$ | $0.90 \pm 0.10$ | $97.6 \pm 14.8$ | $0.90 \pm 0.07$ |
| | MCSL | $49.0 \pm 8.1$ | $0.62 \pm 0.06$ | $54.0 \pm 10.0$ | $0.70 \pm 0.10$ | $53.8 \pm 9.6$ | $0.73 \pm 0.10$ | $57.6 \pm 11.6$ | $0.73 \pm 0.08$ |
| Sep data | PC | $31.2 \pm 5.7$ | $0.30 \pm 0.05$ | $29.0 \pm 5.9$ | $0.39 \pm 0.06$ | $28.5 \pm 6.3$ | $0.44 \pm 0.07$ | $27.9 \pm 6.6$ | $0.47 \pm 0.08$ |
| | GES | $35.1 \pm 8.3$ | $0.48 \pm 0.10$ | $31.6 \pm 9.8$ | $0.57 \pm 0.08$ | $30.0 \pm 8.0$ | $0.62 \pm 0.06$ | $30.5 \pm 10.7$ | $0.64 \pm 0.07$ |
| | DAG-GNN | $29.9 \pm 7.2$ | $0.66 \pm 0.09$ | $20.3 \pm 5.0$ | $0.67 \pm 0.09$ | $18.5 \pm 4.9$ | $0.67 \pm 0.09$ | $18.0 \pm 5.2$ | $0.66 \pm 0.11$ |
| | NOTEARS | $\mathbf{16.3 \pm 3.4}$ | $0.61 \pm 0.08$ | $\mathbf{15.5 \pm 3.2}$ | $0.60 \pm 0.08$ | $15.0 \pm 3.1$ | $0.62 \pm 0.09$ | $15.2 \pm 2.9$ | $0.61 \pm 0.09$ |
| | N-S-MLP | $68.0 \pm 5.4$ | $\mathbf{0.80 \pm 0.04}$ | $22.6 \pm 3.3$ | $\mathbf{0.79 \pm 0.06}$ | $12.7 \pm 2.6$ | $\mathbf{0.80 \pm 0.05}$ | $\mathbf{11.8 \pm 2.8}$ | $\mathbf{0.80 \pm 0.05}$ |
| | MCSL | $32.8, \pm 5.4$ | $0.49 \pm 0.08$ | $26.4 \pm 5.5$ | $0.53 \pm 0.09$ | $23.3 \pm 5.8$ | $0.56 \pm 0.08$ | $23.1 \pm 6.5$ | $0.56 \pm 0.07$ |
| | DS-FCD | $\mathbf{11.6 \pm 5.6}$ | $\mathbf{0.83 \pm 0.11}$ | $\mathbf{7.1 \pm 6.1}$ | $\mathbf{0.90 \pm 0.12}$ | $\mathbf{6.2 \pm 4.7}$ | $\mathbf{0.89 \pm 0.09}$ | $\mathbf{6.0 \pm 5.5}$ | $\mathbf{0.91 \pm 0.11}$ |

Table 16: Results on randomly selecting models-info of partial clients (Non-IID, 20nodes, ER2).

| | | IID | | | | Non-IID | | | |
|---|---|---|---|---|---|---|---|---|---|
| | | ER2 with 10 nodes | | ER2 with 20 nodes | | ER2 with 10 nodes | | ER2 with 20 nodes | |
| | | SHD ↓ | TPR ↑ | SHD ↓ | TPR ↑ | SHD ↓ | TPR ↑ | SHD ↓ | TPR ↑ |
| $\frac{r}{m}$ | 10% | $3.8 \pm 2.4$ | $0.78 \pm 0.14$ | $8.6 \pm 4.8$ | $0.77 \pm 0.13$ | $3.8 \pm 1.4$ | $0.93 \pm 0.05$ | $8.5 \pm 5.4$ | $0.89 \pm 0.07$ |
| | 20% | $3.2 \pm 2.0$ | $0.81 \pm 0.12$ | $6.7 \pm 4.8$ | $0.82 \pm 0.13$ | $2.5 \pm 2.1$ | $0.97 \pm 0.04$ | $8.2 \pm 5.4$ | $0.87 \pm 0.09$ |
| | 50% | $2.9 \pm 1.8$ | $0.83 \pm 0.11$ | $5.8 \pm 4.4$ | $0.85 \pm 0.12$ | $1.8 \pm 1.4$ | $0.99 \pm 0.02$ | $6.3 \pm 5.1$ | $0.89 \pm 0.10$ |
| | 80% | $2.7 \pm 1.9$ | $0.84 \pm 0.12$ | $6.0 \pm 3.9$ | $0.86 \pm 0.10$ | $1.8 \pm 1.3$ | $0.99 \pm 0.02$ | $5.9 \pm 4.1$ | $0.90 \pm 0.08$ |
| | 100% | $2.4 \pm 2.0$ | $0.86 \pm 0.12$ | $6.2 \pm 4.0$ | $0.85 \pm 0.10$ | $1.9 \pm 1.6$ | $0.99 \pm 0.02$ | $6.2 \pm 4.7$ | $0.89 \pm 0.09$ |

and NOTEARS on linear IID data are implemented in the Table 1. DAG-GNN is also a non-linear model. However, the non-linear assumption of DAG-GNN is not the same as the data generation model ANMs assumed in our paper. The second reason comes from its *mechanisms approximation* modules are compulsory to share some parameters. Both NOTEARS-MLP and MCSL have their own advantages. Please refer to Tables 13 and 14, you will find that NOTEARS-MLP performs better when the non-linear functions are MIM and MLP while MCSL works better on GP and GP-add models.

### E.2 Why does our method outperforms other methods even some baseline methods using all data for training?

Let us first discuss the AS-FCD (All-Shared FCD), which shares all model parameters (both $\mathbf{\Phi}$ and $\mathbf{U}$) among all clients. If we set $it_{fl}$ as 1 in AS-FCD, AS-FCD is totally the same as MCSL using all data for training. For simplicity, we mark all parameters (actually $\mathbf{\Phi}$ and $\mathbf{U}$) of client $c_k$ together as $\theta^{c_k}$. Let us consider the $t$-th iteration when all clients receive the average parameters $\theta_t$ from the server and update their parameters by $\theta_t$.

For AS-FCD, firstly, we mark the gradients obtained by using the local data of client $c^k$ for $k \in [m]$ as $g_t^{c_k}$. Then each client $c_k$ updates its parameters for one step by $\theta_t^{c_k} = \theta_t - lr \times g_t^{c_k}$, where $lr$ is the learning rate. Afterwards, the server collects all parameters and averages them to get $\theta_{t+1} = \frac{\sum_{k=1}^{m} \theta_t^{c_k}}{m} = \frac{\sum_{k=1}^{m}(\theta_t - lr \times g_t^{c_k})}{m} = \theta_t - lr \times \frac{\sum_{k=1}^{m} g_t^{c_k}}{m}$. For MCSL, there is only one $\theta$. If MCSL uses full gradient information, then $\theta_{t+1} = \theta_t - lr \times \frac{\sum_{k=1}^{m} g_t^{c_k}}{m}$ (the full gradient is just the average of gradients from all samples). We can find that the updated parameters are totally the same. Then if $it_{fl} > 1$, we average all parameters every $it_{fl}$ iterations. Even though the exact updating procedures are not the same, the expectations of updated parameters are the same. This is why we say that *MCSL trained on all data can serve as an approximate upper bound of our method but unobtainable* in our paper.

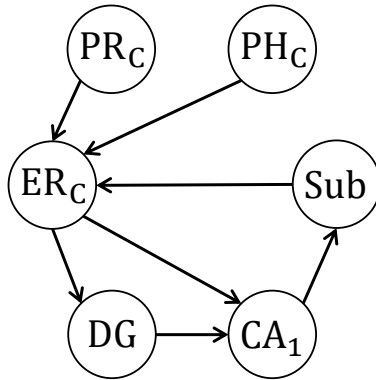

Figure 10: Anatomical causal-effect relationships of **fMRI Hippocampus** dataset

Table 17: Empirical results on **fMRI Hippocampus** dataset (Part 2).

|  | All data | | | Separate data | | | DS-FCD | AS-FCD |
|---|---|---|---|---|---|---|---|---|
|  | GES | N-S-MLP | DAG-GNN | GES | N-S-MLP | DAG-GNN | | |
| SHD ↓ | $8.0 \pm 0.0$ | $9.0 \pm 0.0$ | $5.4 \pm 0.5$ | $8.3 \pm 1.2$ | $11.3 \pm 1.0$ | $8.2 \pm 1.9$ | $\mathbf{6.4 \pm 0.9}$ | $\mathbf{5.0 \pm 0.0}$ |
| NNZ | $11.0 \pm 0.0$ | $12.0 \pm 0.0$ | $3.3 \pm 0.8$ | $8.5 \pm 1.1$ | $14.4 \pm 0.8$ | $5.7 \pm 1.4$ | $6.8 \pm 0.6$ | $5.0 \pm 0.0$ |
| TPR ↑ | $0.43 \pm 0.00$ | $0.43 \pm 0.00$ | $0.23 \pm 0.07$ | $0.31 \pm 0.17$ | $\mathbf{0.44 \pm 0.10}$ | $0.17 \pm 0.18$ | $0.27 \pm 0.12$ | $\mathbf{0.29 \pm 0.00}$ |
| FDR ↓ | $0.73 \pm 0.00$ | $0.75 \pm 0.00$ | $0.52 \pm 0.09$ | $0.75 \pm 0.12$ | $0.78 \pm 0.05$ | $0.80 \pm 0.18$ | $\mathbf{0.72 \pm 0.11}$ | $\mathbf{0.60 \pm 0.00}$ |

In DS-FCD (DAG-Shared FCD) method, only all causal graphs are averaged. However, this partial information-sharing mechanism also helps on benefiting information from other clients to find a better solution (Collins et al., 2021).

## F Discussions on Assumptions

### F.1 Data heterogeneity

The general Non-IID setup should include the distribution shift caused by interventions. Since interventions on some certain variables would also lead to heterogeneous distribution. Previous work (Huang et al., 2020b) has investigated this case and proposes CD-NOD algorithm, which enhances the PC method, to learn from heterogeneous data. However, CD-NOD need to identify some edge directions by capturing the changing information among distributions. That is to say, this method which need to gather all data and cause the raw data leakage, of course. In our paper, we restrict our attention to the ANMs, which care more about the mechanisms and noises shift among different clients. Moreover, finding the identifiability conditions for learning causal graph from the general heterogeneous data (both mechanisms shift and interventional data) in the federated setup is a challenging but important problem, which is left for the future work.

### F.2 Is our Invariant DAG assumption reasonable?

Firstly, let us skip the IID setting of FCD, which only assumes all SEMs are totally the same but data are generated at different local clients. Then, we mainly talk about the Non-IID setting that assumes SEMs vary but DAG is shared among different clients. Essentially, a SEM models the physical processes of a system and the generation process behind observations. (1) Intuitively, different SEMs usually describe different systems. Then, naturally, the DAGs may be different. Following this logic, we would say yes that the invariant DAG assumptions among different SEMs are too strong. (2) The assumption that domain shifts can come from the distribution shifts of the exogenous variables (noise terms in our paper) has been widely accepted, such as IP (Peters et al., 2016) and IRM (Arjovsky et al., 2019). So there is no need to argue this one. Then, let us come back to our FCD. We argue that *different SEMs* is a weak necessary condition of *different systems* but

not the sufficient condition. Because a system can have various SEMs at different statuses (Huang et al., 2020b). Here, we name this kind of system an unstable system. While in Non-IID setting of FCD, we assume that all clients share an unstable system.

In practice, the first example can be fMRI recordings. As pointed in (Huang et al., 2020b), fMRI recordings are usually non-stationary because information flows in the brain may change with stimuli, tasks and attention of the subject. Our federated setting only has one more assumption that fMRI recordings among different clients cannot be shared. The second example can be causal gene regulatory network inference (Omranian et al., 2016). The causal direction among genes, i.e., which gene regulates which gene, is believed to be the same. However, in each individual, the SEM mechanism could vary due to individual properties, such as age, gender, etc.

## G   Data visualization

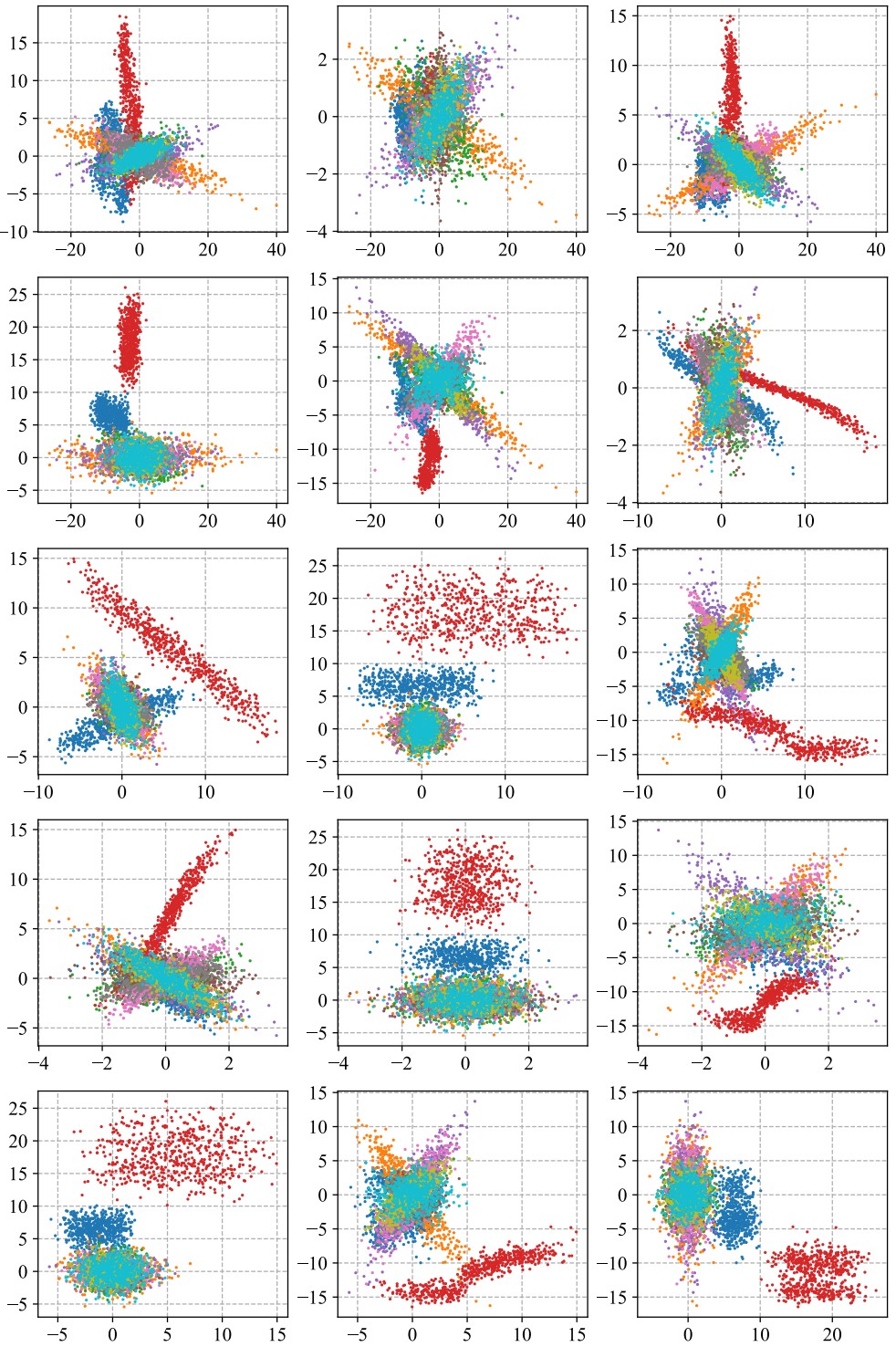

Figure 11: The visualization of simulated Non-IID data with 10 variables, where 6 variables are randomly selected and two of them are chosen for one sub-figure.

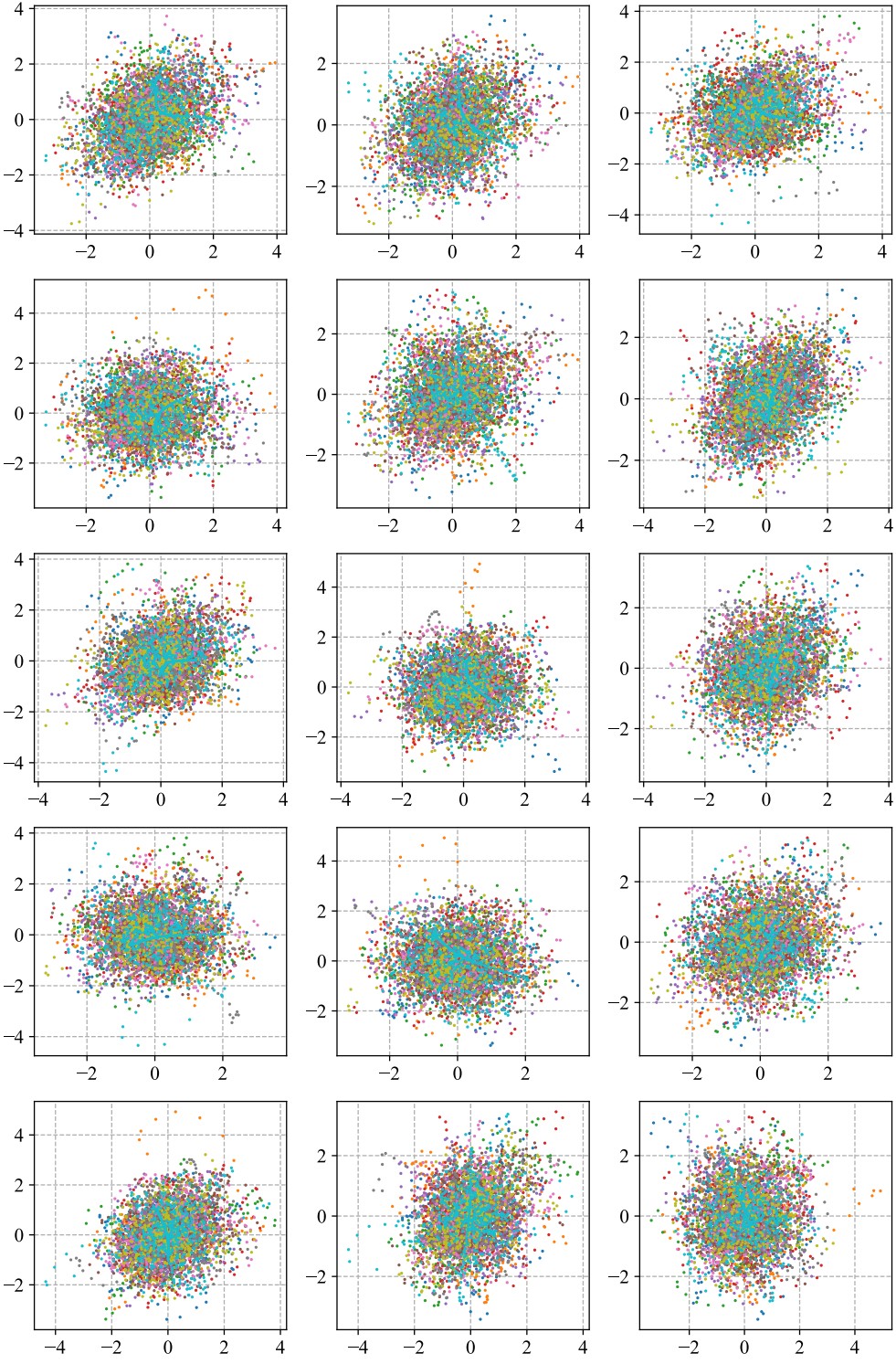

Figure 12: Normalized distribution of real data used in this paper.

