# OpenReview forum: "Federated Causal Discovery with Additive Noise Models"
_TMLR — Rejected by TMLR_

### Review · Reviewer_V5yU · 2022-07-05

**Summary Of Contributions:**

This paper expands on the previous work of NOTEARS (Zheng et al. 2018) and MCSL (Ng et al., 2022b) to develop a new federated causal inference framework. In this setup, observational data are federated across local sites and cannot be centralized.

The causal structure is represented in terms of (1) a directed acyclic graph (DAG); and (2) a set of causal mechanisms (i.e. conditional probabilities) along the arcs of the DAG characterized by a nonlinear function perturbed by independent Gaussian noises. The DAG is assumed to be the same across data silos but the causal mechanisms are different.

This is formulated as a constrained graph learning problem where the constraint is adopted from NOTEARS (Zheng et al. 2018) & MCSL (Ng et al. 2022b) to enforced that the learned graph is a DAG. This allows the DAG to be learned by a direct application of FedAvg which proceeds in multiple rounds of communications & aggregations. Within each iteration, each data client is also allowed to learn a local, customized set of conditional probabilities (causal mechanisms) of the factorization induced by the previous estimate of the graph.

The authors acknowledge that up to the formulation, their work is identical to a concurrent work of NOTEARS-ADMM (Ng & Zhang, 2022). The solution techniques are different and empirical results show that the proposed method has better performance in settings with heterogeneous data. Given this, I would view this work as a practical contribution that expands and refines prior work to a new empirical setting (i.e. heterogeneous data distribution)

**Broader Impact Concerns:**

It is currently unclear whether sharing the causality structure between clients with private data could lead to leakage of private information of a client. It would be good if the paper can discuss this in a broader impact statement, discussing in whether there are anticipated ways in which the proposed framework might be misused.

**Requested Changes:**

Going by the above weaknesses, I would suggest the following changes:

1. Please elaborate if there are any differences in the problem setup between the proposed approach & (personalized) federated graph learning? If not, there should be positioning with respect to the respective prior work, some of which has been pointed out above. Comparisons with those works are also necessary if the relevance scope of this paper indeed needs to be expanded.

 2. Going together, the key assumptions of DAG invariant & that the DAG invariant can be uniquely identified by a single data source seem to make this problem setup no different than a generic federated graph learning task; and more importantly, remove the need of a federated framework -- in the end, if the DAG variant can be identified from a single source, we can simply learn it first using previous centralized causal approaches & then broadcast it to others so that they can customize the causal mechanisms -- the authors need to either make this clear how such assumptions are still practically relevant to federated learning or consider rework the framework under more relaxed assumptions

3. The no confounder assumption does not seem to be practical in federated setting as I mentioned above. Please discuss if you disagree or consider rework the method to relax it.

4. Please consider moving important discussions, loss definitions and extra assumptions in the appendix back to the main text.

5. Section 4.3 needs to be expanded with more elaboration.

**Strengths And Weaknesses:**

Strength:

1. The problem considered in this paper is interesting and relatively new which has only been investigated by a concurrent work that the authors pointed out. I also agree with the authors' motivation that given stricter data regulation such as the GDPR, it is possible that in many applications when data cannot be centralized, classical causal inference approach is no longer viable.

2. The authors have conducted extensive experiments to demonstrate the practical improvement of the proposed approach in comparison to baselines within the literature review of this paper.

3. Other than a few minor grammar issues, the paper is mostly clear and easy to understand. I also appreciate the authors' effort in preparing an extensive appendix addressing many questions that might arise from the content in the main text, as well as providing more results & comparison.

Weaknesses:

1. Despite the contribution claim on causal inference, the proposed solution was developed under assumptions that make it unclear what is the line between it & the broader class of (personalized) federated graph learning. For example:

https://federated-learning.org/fl-ijcai-2021/FTL-IJCAI21_paper_20.pdf
https://arxiv.org/abs/2104.07145
https://arxiv.org/pdf/2203.00829.pdf

Going by the technical content in the main text, I do not see any aspects other than the DAG constraint which are specific to causal inference. Given this, I think the causal aspect of this paper is minimal & it should have been positioned in the broader scope of (personalized) federated graph learning, which is currently not well covered.

2. Each local data distribution can admit multiple factorizations so there is always the issue of identifiability (i.e. among those admitted, which one represents the true causal structure?) if we only learn with passive observations.

The authors did point out another set of assumptions under which the true causal structure can be uniquely identified even with passive observations. However, while the argument is theoretically sound, it is practically irrelevant: if I read correctly, the assumptions indicate that there exists at least one local data distribution that uniquely identifies the true causal graph but if that is the case, we do not need this entire federated learning mechanism.

3. I also feel that the assumption of invariant & unique DAG factorization (3.1, B.5 & B.6) among local data silos is what blur the line between causal inference and graph learning, as I mentioned in point 1 above. To me, this work should be either positioned in the generic graph learning context or specialized into causal inference but relaxed the above assumptions.

4. Going together, the assumptions of invariant DAG & no confounder is somewhat strong in the heterogeneous data context. It is not uncommon for different data silos to be collected under different contexts, which might impact the causal structure. So ignoring it does not seem like a practical choice.

5. Several important parts & discussions were deferred to the appendix but in my opinion, those should have been surfaced in the main text for better clarity, especially the extra assumptions (B5 & B6) regarding identifiability & the score function in Appendix A.

6. Section 4.3 seems very adhoc. How do we know which threshold is best to use? It is not clear what "iterative thresholding to cut off the edge with minimum weight until the graph is acyclic" means?

---

> ### Author Response · Authors · 2022-07-19
> **Response to Reviewer V5yU [Q4-6]**
>
> **[Q4. Paraphrasing.]** Thanks! We have moved some contents from the Appendix to the main text according to your suggestion.
>
> **[Q5. Adhoc thresholding.]** (1) For continuous optimization, we leverage Gumbel-Sigmoid to approximate the binary mask. That is to say, the exact $0$ or $1$ is hard to get. (2) The other issue is raised by ALM, which finally converts the hard constraint optimization to a soft-constraint optimization (even with a large penalty coefficient). Therefore, many positions with small values would be obtained. In our method, since all mask values are in $[0, 1]$, we just take the middle value $0.5$ as the threshold to prune the edges, which follows the same way in our baseline method MCSL (Ng et al., SDM2022). The iterative thresholding method is also taken to deal with the case that the learned graph is cyclic. This may happen if the number of variables is large ($40$ variables in our paper). Because, in numerical optimization, the constraint penalty exponentially decreases with the number of variables. To deal with the cyclic graph, we one-by-one cut edge with the minimum value until we have a DAG. To our knowledge, until now, all continuous search methods for causal discovery suffer from these two problems. It is an interesting future direction to be investigated.
>
> **[Q6. Broader Impact concerns.]** In FL, the server and some clients participate in this process. While as we talked about above, the DAG is shared among all clients. The FCD is motivated by "data on each client is not enough for identifying up the ground-truth DAG". That is to say, the causal graph information is not private for clients. For the server, it depends. In our previous motivations, we actually only care about the "raw data leakage" problem but did not take the privacy of the causal graph into consideration.
> In real-world scenarios, some of the causal relations can be public such as in disease research. For these cases, our method can still work. However, as you suggest, causality structure sometimes may also be private information. This problem can also be easily solved. In FCD, the only rule of the server is to (1) aggregate and average the CGLs and (2) broadcast the new CGL to clients for updating their local CGLs. Therefore, the server can be easily replaced by selecting a client to do the same thing. We name this client the proxy server. For the proxy server, it needs to play two roles, including training its own model and taking the server's duties. Then, in the communication round, other clients communicate with the proxy server instead of a real server.
>
> [1] Differentiable Causal Discovery from Interventional Data. NeurIPS2020.
> [2] MissDAG: Causal Discovery in the Presence of Missing Data with Continuous Additive Noise Models. arXiv2022.
> [3] DYNOTEARS: Structure Learning from Time-Series Data. AISTATS2020.
> [4] A Review of Causal Discovery Algorithms Based on Graphical Models. Frontiers in Genetics 2019.

---

> ### Author Response · Authors · 2022-07-19
> **Response to Reviewer V5yU [Q1-3]**
>
> Thanks for your review. We point-wisely address all your concerns as below and have updated some of the discussions in the revision.
>
> **[Q1. The difference with federated graph learning.]** Federation is a setup for learning graphs from decentralized data. Since we guess that the problem you concern with is the difference between causal discovery (CD) and graph neural network (GNN) learning. There are four main reasons, which make CD and GNN two different research lines. (1) Nodes in a causal graph represent variables and directed edges describe the cause-effect relation between different variables. In GNN, the graph talks more about the graph-type data, such as social networks, protein networks, and traffic networks. (2) Networks in CD are leveraged to learn the causal mechanisms while networks in GNN are taken to achieve node embedding and feature extraction. (3) Learned causal graph can be taken for interventional and counterfactual reasoning. (4) CD cares more about identifiability. That is to say, it is important to exactly identify the true causal process underlying the observations. These discussions have been updated in the revision.
>
> **[Q2. Identifiability.]** (1) The identifiability is a theoretical property with the assumption of infinite data. But, in practice, this requirement is hard to meet. (2) We agreed with you that federated learning is useless if one client can have enough samples to identify the true DAG. This can be achieved by your suggestion. (*we can simply learn it first using previous centralized causal approaches & then broadcast it to others so that they can customize the causal mechanisms.*)
> However, in real scenarios, each client only owns samples of finite number, which leads to the data variance problem when only learning with local data. From our experimental results (reported as -Sep), we can find that the learning results only on local data are unsatisfactory. Therefore, our strongest motivation is exchanging the training information among clients to co-learn a better causal graph. The motivation has been well clarified in our revised paper.
>
> **[Q3. No confounder assumption.]** Agreed with that latent confounder is common in many practical scenarios. Handling latent confounders is a fundamentally important but really hard problem in the traditional continuous causal discovery not to mention the federated setup. Until now, the theoretical results on the identifiability of causal discovery with latent confounders are always too weak to be used in practice since too strict assumptions are taken. Identifying the ground-truth DAGs is far away from being solved. Based on this situation, we do think that the first step of designing causal discovery with latent variables from the data in one domain should be taken before we consider it in the federated setup.
> Moreover, as we said, our method can serve as a framework for causal discovery in the federated setup. It can incorporate advanced methods to deal with interventional data [1], incomplete data [2], and time-series data [3]. That is to say, we can also consider incorporating the advanced continuous methods into our framework if they can well handle latent variables from the data in one domain. Therefore, we leave these combinations as future work in the manuscript. In the revision, we have updated the discussion of our method as a framework for causal discovery in the federated setup.

---

### Review · Reviewer_vCrT · 2022-07-05

**Summary Of Contributions:**

The authors propose a federated learning algorithm to implement a score-based causal discovery algorithm that is designed for additive noise model SCMs.

**Broader Impact Concerns:**

I have no broad impact concerns.

**Requested Changes:**

The writing and specifically certain claims get subjective at times. For example, "This is indeed good news for users but a definite disaster for many companies."

"A causal model is defined as a triple" Please resort to Pearl 2009 for a formal definition. This definition is missing at the very least the product probability measure over the exogenous variables.

The writing needs improvement. For example, the following sentence is missing the latter half of it:
"While NOTEARS only solves the linear case, which assumes that all the causal mechanisms are linear and can be learned by a weighted matrix"

What is t in equation 7? Do you mean k?

I wasn't able to find a consistency result in the paper. Namely, is it clear that by optimizing the combined loss function at each client and aggregating the result that we can find the true graph? The source of my concern is that authors suggest optimizing equation (6) or (7) is sufficient for finding the true graph. Especially that having different distributions at each client is allowed, this is not very clear.

I am not sure how the proposed method can handle nonIID data. Even the likelihood maximization is unclear with nonIID data. My guess is that authors meant that data can arise from different distributions in different clients. But this is not the same as nonIID assumtion. If so, this needs to be fixed in the manuscript before acceptance as calling this setting nonIID might be very misleading.

A verbal depiction of the overall algorithm would be very helpful for clear exposition. It seems that each client receives only the current parameters \alpha, \rho and finds (best it can through gradient descent) the best graph/causal mechanism. Server somehow combines these and returns combined graph in the form of a new U matrix and repeats the process.

One thing that seems inconsistent in the algorithms is that U^{i,c_k} should be used in the definition of SP^{c_k} rather than U, I think. Please check.

Another thing that concerns me is that how do we know that there will not be any cyclic behavior? How do we guarantee convergence through this method? For example, suppose we have two clients only. But although they have same graphs, their distributions are different so that after each update, we cycle back and forth between two U values. Why can't this happen?

**Strengths And Weaknesses:**

The topic is new and not addressed - as far as I am aware of. Although I haven't yet seen this as a practical problem it is foreseeable that it might become relevant in near future. There are several synthetic and one real data experiment.

The proposed algorithm is missing the consistency result. Some claims (e.g. nonIID setting) are not substantiated.

---

> ### Author Response · Authors · 2022-07-19
> **Response to Reviewer vCrT [Q5-8]**
>
> **[Q5. Non-IID assumption.]** Thanks for the great suggestion. Our NonIID setup is not the general case but restricts the change of data distributions on each client by (1) mechanism or (2) noise shift. In our revision, this has been well clarified and we also change our statement of *Non-IID data* to *heterogeneous data*. To deal with the data heterogeneity raised by our assumption, our method learns the personalized CMA model on each client to deal with the causal mechanisms shift. The other change is caused by noise scale shifts. In our method, instead of directly modeling the likelihood of observations, we choose to model the noise distribution by density transformation. For the noises shift, we carry with the Gaussian distributions with different scales, which can still be equally modeled by $\ell_2$ loss.
>
> **[Q6. Verbal depiction of the overall algorithm.]** Thanks for the great suggestion. Your description of our method is quite accurate, and we have expanded it to the following explanation and added it to the revised paper. For each server update procedure, the server receives $U^{c_k}$ from clients and averages these masks to get $U^{new}$. Meanwhile, $h$ loss is calculated on the server to update the Lagrangian coefficients $\alpha^{new}$ and $\rho^{new}$. Then, $U^{new}$, $\alpha^{new}$, and $\rho^{new}$ are sent to clients for local training. We have clarified these in the revision.
>
> **[Q7. Inconsistent description.]** Thanks! We have corrected them to be consistent.
>
> **[Q8. Cyclic behavior.]** The cyclic behavior is avoided by the synchronous local training of all clients. During the local updating procedures, all clients are independently trained at the same time. Then, the server receives parameters from clients, averages these $U$s, and sends $U_{new}$ to all clients. For local data belonging to different distributions, the optimal binary adjacency matrix is totally the same among all clients. But the learned causal mechanisms are different. In the revision, we have updated some visualizations of the learning procedures, including the changes of the learned graph, $h$ loss, penalty coefficients, and the $\ell_1$ and SHD distances between graphs learned on two different clients.
>
> [1] Optimal Structure Identification With Greedy Search. JMLR2002.
> [2] Causal Discovery with Continuous Additive Noise Models. JMLR2014.
> [3] Globally optimal score-based learning of directed acyclic graphs in high-dimensions. NeurIPS2019.

---

> > ### Comment · Reviewer_vCrT · 2022-08-05
> > **Response to authors**
> >
> > Hello,
> >
> > Thank you for the responses! Here are some further remarks on a few points:
> >
> > - I don't think the answer to my consistency question addresses the concerns. I am not concerned about the identifiability results in the ANM literature. The question was more about the algorithm that optimizes the function locally with "possibly different distributions". Identifiability results do not directly imply the proposed algorithmic approach is consistent with multiple agents accessing different datasets. This is not addressed by the authors.
> >
> > - the addressing of the potential cyclic behavior should also be more rigorous. Perhaps you would like to say something along the lines of "loss function is monotonically decreasing with each update". Something of this sort to make it a rigorous claim that there will not be a cyclic behavior is needed.

---

> > > ### Author Response · Authors · 2022-08-09
> > > **Covergence results of the optimization**
> > >
> > > Thanks for your time and great efforts. We get that your concerns mainly come from (1) Does the global maximum of Equations (6) and (7) correspond to the ground-truth DAG? (2) Can DS-FCD guarantee to reach the global maximum of Equations (6) and (7) ?  and (3) Can the output of DS-FCD converge to a stationary point?
> > >
> > > (1) The results are mainly based on the consistency of the BIC score for identifying the causal graph in the standard setting instead of the federated setup. $\langle 1 \rangle$ For linear models, the consistency theory in the standard has been well developed [1,2]. That is to say, the ground-truth DAG corresponds to the global maximum of the score on each client. Therefore, the ground-truth DAG also corresponds to the global maximum of Equations (6) and (7) since the overall score is just the sum of scores on all clients. $\langle 2 \rangle$ However, for non-linear ANMs, even though many practical methods, e.g., MCSL, NOTEARS-MLP, and CD-RL, have been proposed to solve this problem by maximizing the BIC score, the theoretical results of consistency are still lacking and would be an interesting future work to be investigated. This discussion has been updated in Appendix C.5.
> > >
> > > (2) Unfortunately, **the global maximum cannot be guaranteed to reach by taking our gradient-based DAG-Shared FCD method**. The main reasons come from *the non-convexity hard constraint*, which makes the overall learning objective a non-convex function.
> > >
> > > (3) The same as NOTEARS and its following works, **our method can reach a stationary point** instead of the global maximum (the ground-truth DAG). The general case in our paper is the heterogeneous data with different distributions on each client. Based on the theoretical results developed in the FL on Non-IID data [3,4,5], we have established the convergence results of our FCD method in Appendix C.9.2 for heterogeneous data. With the increase of the optimization iteration $T$, the gradients can be bounded. This also can be an explanation for the no cyclic behavior.
> > >
> > > Let us know if you have any further questions.
> > >
> > > [1] High-Dimensional Learning of Linear Causal Networks via Inverse Covariance Estimation. JMLR2014.
> > > [2] Globally optimal score-based learning of directed acyclic graphs in high-dimensions. NeurIPS2019.
> > > [3] Federated Learning with Personalization Layers. AISTATS2020.
> > > [4] On the Convergence of Local Descent Methods in Federated Learning. arXiv2019.
> > > [5] Federated Learning with Partial Model Personalization. ICML2022.

---

> ### Author Response · Authors · 2022-07-19
> **Response to Reviewer vCrT [Q1-4]**
>
> Thanks for your constructive comments. We point-wisely address all your concerns as below and have updated our paper according to your suggestions.
>
> **[Q1. Writing improvement.]** We have improved the writing quality and made our statement more objective.
>
> **[Q2. Formal definition.]** Thanks. We limit our expression to the structural causal model. Our SCM formulation follows the definition in *Causal Inference in Statistics: A Primer* (See Chapter 1.5.1 and examples in Chapter 2. Pearl 2016). In our SCM, all noise terms serve as the exogenous variables and all observed variables are included in the endogenous set. In the intact causal graph of ANMs, we just fix directed edges from $\epsilon_i$ to $X_i$ and assume the distribution of $\epsilon_i$. Therefore, in our paper, we focus on learning the causal relations among all observed variables. We have polished the formulation to better reflect the exogenous and endogenous variables.
>
> **[Q3. Equation 7.]** $t$ in Eq. (7) is not $k$. We take ALM to solve the hard-constrained optimization problem in Eq. (6) by a series of unconstrained sub-problems. $t$ is the index of $t$-th sub-problem. We have improved our presentation to avoid confusion.
>
> **[Q4. Consistency results need some discussion.]** Thanks for your suggestions.
> Actually, for linear additive noise models with Gaussian noises, the consistent results for maximizing the BIC score to identify the causal graph (MEC or DAG) have been well established [1,2]. For this case, with the DAG space constraint, the unique maximum of score function $\mathcal{S}^{c_k} (\mathcal{D}^{c_k}, \Phi^{c_k}, U^{c_k})$ with BIC score corresponds to the ground-truth DAG. Even for the high-dimensional consistency for linear Gaussian SEM in the case when the model is identifiable [3]. Since the ground-truth $\mathcal{G}$ corresponds to each $\mathcal{S}^{c_k}$, the global maximum $arg \max_{ \Phi,  U}\sum_{k=1}^{m} \ \mathcal{S}^{c_k} (\mathcal{D}^{c_k},  \Phi^{c_k},  U)$ with DAG constraint can lead to the ground-truth causal graph. For nonlinear ANMs, however, even though many practical methods, e.g., MCSL, NOTEARS-MLP, and CD-RL, have been proposed to solve this problem by maximizing the BIC score, the theoretical results of consistency are still lacking and would be an interesting future work to be investigated. Therefore, our framework based on these methods inherits the theoretical limit for the nonlinear case. From our paper, however, empirical results can still show the effectiveness of the method. The discussion has been updated in our revision. The other problem is optimization. Although the smooth characterization of DAGs is exact, it also inherits the hard non-convexity of the DAG space. The non-convex DAG constraint term makes the overall problem a non-convex optimization, which cannot be exactly solved by gradient-based methods. By relying solely on the gradient information, ALM can only lead to stationary points instead of the global optimum. Please refer to the Appendix (Discussions on Our Method) for more details.

---

### Review · Reviewer_o1PK · 2022-07-10

**Summary Of Contributions:**

This work proposes federated causal discovery to address privacy issues in causal discovery. Their method aims to protect user privacy for gradient based causal discovery methods. Results on both synthetic and real-world datasets show the proposed method is more accurate in the task of causal discovery.


**Requested Changes:**

1. In the first contribution, it is very vague to me (not an expert in FL) why FL can only deal with conditional distribution and why only fitting conditional distribution is not good enough for causal discovery.

2. In the second contribution, the authors claim that in causal discovery heterogeneity always means the distribution shift of all variables. This is not true from what I understood. For example, the joint distribution becomes different when a certain conditional distribution changes if you decompose the joint distribution according to the group-truth causal graph.

3. This work defines non-i.i.d. as data generated by the same causal model but with different causal mechanisms. But it is more regular to find non-i.i.d. data generated by interventions on a certain variable, see [1,2]. I guess the authors need some detailed justification for their setting.

4. The claim right below Assumption 3.1 is not quite accurate. It is possible that with the different causal graphs you still end up with the same factorization. That's why constraint-based methods can only provide a Markovian equivalent class as the output.

5. Ideally, federated causal discovery should be able to take any gradient-based causal discovery method as its input and get the job done, however, this work only considers a special case.

6. Is it possible to show how well the proposed method works in terms of privacy protection? Since just comparing the performance of causal discovery cannot fully support the motivation of this work.

[1] Koyama, Masanori, and Shoichiro Yamaguchi. "Out-of-distribution generalization with maximal invariant predictor." (2020).
[2] Zhang, Kun, Mingming Gong, Petar Stojanov, Biwei Huang, Qingsong Liu, and Clark Glymour. "Domain adaptation as a problem of inference on graphical models." Advances in Neural Information Processing Systems 33 (2020): 4965-4976.

**Strengths And Weaknesses:**

Strength:
1. The work investigates a new problem: causal discovery under the FL setting.
2. The paper is well presented, it is clear what the authors claim and what's special about the methodology.
3. Experiments are done across different datasets and results are convincing in terms of accuracy of the proposed algorithm.

Weakness:
1. The experiments show nothing about privacy protection.
2. There are some technically inaccurate/controversial descriptions in the paper, see requested changes.

---

> ### Author Response · Authors · 2022-07-19
> **Response to Reviewer o1PK [Q1-6]**
>
> Thanks for your constructive comments. We point-wisely address all your concerns as below and have updated our paper according to your suggestions.
>
> **[Q1. Conditional distribution.]** We mean that *Previous FL tasks* mainly focus on estimating the conditional distribution $P(Y|X)$ in supervised learning tasks, e.g., image classification [1], sequence tagging [2], and feature prediction [3]. For federated causal discovery, which is an unsupervised learning task, our method directly models the joint distribution of all observed variables since it leverages the score-based methods, which usually include the maximum likelihood of all observations. Here, we illustrate it as the **different modeling styles** between FCD and traditional FL.
>
> P.S. The joint distribution of all variables in a causal graph can relate to the factorization of conditional distributions according to Eq. (2) in our paper.
>
> **[Q2. Heterogeneity/distribution shift.]** Thanks for pointing it out. What we wanted to express was that our method can handle possible distribution shifts of all variables. Our previous statement was confusing and we have changed our expression to *FCD handles a generative model where data heterogeneity can admit the joint distribution shift of all variables* in the revision.
>
> **[Q3. Non-IID data.]** The *mechanism change* of the causal model was proposed and defined as the change of $P(X_i|PA_i)$ for adapting to the dynamic aspects of the environment (See more details in Section 2 and Definition 1 in [4]). According to this definition, interventions can be seen as a special case of these environmental changes, where the external influence involves fixing a designated variable to some predetermined value. Actually, in general, the external influence may be milder, merely changing the conditional probability of a variable, given its causes [4,5]. In our paper, we restrict our attention to the ANMs, where the heterogeneity is defined to be caused by the functions and noise shift among different clients. Following your suggestions, we have clarified our setting and changed the statement of *Non-IID data* to *heterogeneous data* for the exact clarification in the main text. Moreover, finding the identifiability conditions for learning causal graphs from the general heterogeneous data (both mechanisms shift and interventional data) in the federated setup is a challenging but important problem, which is left for future work. We have updated the discussions in Appendix F.1.
>
> **[Q4. Claims below Assumption 3.1.]** We respectively disagree that different causal graphs could have the same factorization. The decomposition of the joint distribution according to a causal graph is unique, as defined in Eq (2). However, if one aims to learn causal graphs from observational data only, one may suffer from the non-identifiability issue, i.e., different DAGs and the corresponding factorizations generate the same observational distribution and these DAGs are in the MEC. We have rephrased our statements to incorporate the above explanations.
>
> **[Q5. More gradient-based causal discovery methods.]** Not exactly. In practice, all gradient-based methods can be incorporated into our AS-FCD framework to deal with the homogeneous/IID data. However, to deal with the heterogeneous data, we prefer that the baseline methods can separately learn the causal graph and causal mechanisms. Unfortunately, many works are not in this fashion, such as GraN-DAG (Lachapelle et al., ICLR2020), CD-RL (Zhu et al., ICLR2020), and their following works. We have added the discussions in the revised paper.
>
> **[Q6. Privacy protection?]** The aim of our work, and federated learning in general, is not to provide a full solution to privacy protection. Instead, it is the first step towards this goal, i.e., no sharing of data between clients. To further protect privacy, more constraints need to be added to the federated learning framework, such as the prevention of information leakage from gradient sharing, which is studied under the privacy umbrella. To further enhance privacy protection, our method can also include more advanced privacy protection techniques, which we will consider in future work [6].
>
> [1] Communication-Efficient Learning of Deep Networks from Decentralized Data, AISTATS2017.
> [2] FedNLP: Benchmarking Federated Learning Methods for Natural Language Processing Tasks. NAACL2022.
> [3] Advances and Open Problems in Federated Learning. arXiv2019.
> [4] Causal Discovery from Changes. UAI2001.
> [5] Causal Discovery from Heterogeneous/Nonstationary Data. JMLR2020.
> [6] Federated Learning and Differential Privacy: Software tools analysis, the Sherpa.ai FL framework and methodological guidelines for preserving data privacy. arXiv2020.

---

### Comment · Action_Editors · 2022-07-11
**Rebuttal**

Dear Reviewers, Thank you for the timely and insightful reviews!

We will now start the discussion period.

The Authors are encouraged to respond to the reviews.

The goal of the discussion period is for the Reviewers to gather all the information needed to be comfortable submitting a decision recommendation for this submission within 2 weeks.

---

### Decision · Action_Editors · 2022-08-30

**Recommendation:** Reject

**Comment:**

The manuscript proposes a federated learning framework for learning the structure of directed acyclic graphs (DAGs, also known as Bayesian networks) from decentralized data. The framework has a two-level structure consisting of causal graph learning (CGL) part and causal mechanisms approximating (CMA) part. Only CGL parts of clients are shared via an averaging strategy during federated learning and CMA parts are updated locally to better handle data heterogeneity across clients. Experimental results are provided on synthetic Gaussian additive noise models data and on learning anatomical cause-effect relationships of fMRI Hippocampus dataset.

Reviewers acknowledged several strengths of the manuscript including a causal discovery under the federated learning setting as a new problem which might become relevant in the near future. There are however several major concerns:
1. The work should be positioned clearly that it is an adaptation of federated learning for structure learning instead of causal inference. Latent confounder and identifiability issues are not taken into account in the manuscript; these are the challenges that make causal inference uniquely difficult in a federated setting.
2. When assessing the contribution in terms of federated learning for structure learning, the proposed method lacks justification on why it is preferable to prior work in this direction.
3. In the experimental results, only results on structure learning are shown. There is no result related to privacy protection. During the discussion period, the authors acknowledged that causal structure can reveal sensitive information and further suggested using one participating client to be the proxy server as the solution. This needs clarifications as the causal information would still be revealed to this particular actor, so the issue still exits.

Based on the reviews, we agreed that the setup considered in the manuscript does spark up interesting challenges that are worth investigating but the manuscript has not really addressed those.